# Comparative Neuroanatomy of Pediveliger Larvae of Various Bivalves from the Sea of Japan

**DOI:** 10.3390/biology12101341

**Published:** 2023-10-17

**Authors:** Viktoriya Nikishchenko, Nataliya Kolotukhina, Vyacheslav Dyachuk

**Affiliations:** A.V. Zhirmunsky National Scientific Center of Marine Biology, Far Eastern Branch, Russian Academy of Sciences, 690041 Vladivostok, Russia; niktori2000@gmail.com (V.N.); kolotukhina.nata@mail.ru (N.K.)

**Keywords:** neuroanatomy, larvae, ecological niches, serotonin, FMRFamide, apical organ, ganglia, bivalves

## Abstract

**Simple Summary:**

Here, we described and compared the morphology of the nervous system of the late larval stage of bivalve mollusks, pediveligers, using confocal microscopy. Pediveligers from four ecological groups were studied: burrowing (*Callista brevisiphonata*, *Mactromeris polynyma*), cementing (*Pododesmus macrochisma*, *Crassostrea gigas*), attaching to byssus (*Kellia japonica*, *Crenomytilus grayanus*) and mobile forms (*Mizuhopecten yessoensis*, *Azumapecten farreri*). We tried to identify possible similarities and/or differences in a number of structures of the nervous system in pediveligers of different species and correlate them with the lifestyle of a particular ecological group. We have not found a direct connection between the presence or absence of a number of peripheral neurostructures with an ecological group, but we have identified a possible evolutionary connection. A stomatogastric neuron communicating with the apical organ was found in species from the Heterodonta clade. The enteric nervous system and unrelated to the apical organ neurons present in Pectinida were also detected.

**Abstract:**

Here, we describe the nervous system structures from pediveligers of eight bivalve species (*Callista brevisiphonata*, *Mactromeris polynyma*, *Crenomytilus grayanus*, *Kellia japonica*, *Mizuhopecten yessoensis*, and *Azumapecten farreri*) with different modes of life in their adult stages, corresponding to the ecological niches that they occupy (burrowing, cemented, byssally attached, and mobile forms). We have identified neuromorphological features of the central and peripheral nervous systems in larval bivalves. We show that the unpaired sensory apical organ is still present in pediveligers along with the developing paired cerebral ganglia characteristic of an adult mollusk. Pediveligers have the pleural ganglia connected to the pedal ganglia via the pedal nerve cords and to the visceral ganglia via the lateral nerve cords. We have found a number of structures of the peripheral nervous system whose presence varies between pediveligers of different species. Mactromeris, Callista, and Pododesmus have 5-HT-immunopositive stomatogastric neurons, whereas the Yesso and Farrer’s scallops have an FMRFamide-immunopositive enteric nervous system. The innervation of the anterior part of the velum is connected to a system of the apical organ and cerebral ganglia, and the innervation of the posterior part is connected to the visceral ganglia. Most differences in the structure of the peripheral elements of the nervous system are species-specific and weakly depend on the ecological niche that pediveligers occupy.

## 1. Introduction

Bivalve mollusks (Bivalvia) comprise an extensive class in the phylum Mollusca. It numbers a total of about 10,000 species known to date [1] that lead exclusively aquatic lives predominantly in benthic environments. Bivalves can occupy various ecological niches at the bottom of aquatic environments, which eventually determines their morphology and behavioral strategy in these habitat conditions that are characteristic of certain species. Depending on the substrate occupied by bivalves and their mode of locomotion and attachment, several ecologic groups of bivalves are distinguished: byssally attached (*Mytilus*, *Lima*), cemented (*Ostrea*, *Pododesmus*), reclining (*Anadara*, *Placuna*), swimming (some species of Pectinidae), burrowing into sediments (*Mya*, *Solen*), boring (*Pholas*, *Teredo*), and nesting (*Hiatella*) [2]. It is worth noting that the categories of this classification do not have clear-cut boundaries, because the lives of some species include features of different ecological groups [3].

Most bivalve species have ciliated pelagic trochophore and veliger larvae in their life history [4,5,6,7,8,9,10]. The patterns of behavior and the rate of development of trochophore-type larvae are largely determined by the presence of locomotive organs (cilia and muscles) in larvae. The larval nervous system differs greatly in the early stages of development in different groups. However, common mechanisms that control its formation have been described [9,10].

The adult bivalve central nervous system (CNS) is bilaterally symmetrical, represented by three pairs of ganglia: the cerebro-pleural (cerebral), pedal, and visceral [10,11,12,13,14]. The cerebro-pleural ganglion (CPG) is the result of fusion of the paired cerebral (CG) and pleural (PlG) ganglia at the last stages of larval development [11,15]. The fused pedal ganglion (PG) is connected to the cerebro-pleural ganglion via paired cerebro-pleuro-pedal connectives and to the visceral ganglion (VG) via paired cerebro-pleuro-visceral connectives [10,12,14,16]. The pairs of CPG and VG are located on the visceral nerve cords, while the PG is located in the ventral nerve cords [14]. Some bivalve species are also distinguished by having accessory ganglia (AcG), which are paired formations underlying the VG [17,18]. These are connected to the optic nerves of the mantle eyes and are involved in photoreception in adult mollusks [19].

The morphologies of all ganglia and the connectives and commissures associated with them vary at the species level, presumably indicating the adaptation to a particular mode of life characteristic of a certain species. Mollusks leading a slow-moving life (*Anadara*) are reported to have a shorter visceral commissure and a tendency of the pairs of VG to fuse, while in burrowing forms (*Glycymeris*, *Mactra*, and *Mercenaria*), the PG fuse into a single node, and the number of pedal nerves extending from them increases, which may indicate a pronounced development of the foot. Mobile or swimming scallops have the VG fused and larger in size, with their connectives shortened, which is a clear illustration of the complex organization of their CNS [20,21]. In general, bivalves show a tendency toward decentralization of their nervous system, as evidenced by the reduction of the CG and its fusion with the PlG, as well as by the development of the VG. The latter is manifested as a tendency of the paired structures of this ganglion to fuse and an increase in the number of nerves innervating various parts of the animal’s body.

The morphology and physiology of development of planktotrophic bivalve larvae have been studied quite comprehensively, with larval cultures successfully grown to the late veliger and even pediveliger stages in laboratory conditions [7,9,10,22]. However, neither studies of pre- and post-metamorphosis (late) development stages nor investigations into the cellular mechanisms of rearrangement of the larval mode of life in the juvenile period have been published to date.

We aimed to carry out a comparative study of the nervous system structure in pediveliger larvae of different bivalve species with different ecological niches from Peter the Great Bay. Pediveliger is the last larval stage of development of bivalves before becoming the benthic juvenile form of an adult animal. At this stage, the larva undergoes metamorphosis and the formation of a structural plan characteristic of an adult mollusk. The species of bivalves were selected purposefully, in accordance with the ecological niches they occupy. Thus, in our study, we compared the neural morphologies of burrowing (*Callista brevisiphonata*, *Mactromeris polynyma*), swimming (*Mizuhopeten yessoensis*, *Azumapecten farreri*), cemented (*Pododesmus macrochisma*, *Crassostrea gigas*), and byssally attached (*Crenomytilus grayanus*, *Kellia japonica*) species of bivalve mollusks.

## 2. Materials and Methods

### 2.1. Larvae

The objects of the study were larval bivalves from plankton samples taken off Cape Pashinnikov, Vostok Bay, Sea of Japan (42°53′34.9″ N, 132°44′18.9″ E). Larvae were collected from the depth layer of 0–10 m with Juday plankton nets (mesh size of 100 and 150 μm) during the period from June to September 2020–2022. The samples contained various larval developmental stages of bivalves—D-veligers, late veligers, and pediveligers—of which only the late stages (pediveligers) were selected for further study, according to morphological keys published earlier [23,24] and to previous descriptions of cell morphology [25,26,27,28,29,30]. The main morphological characteritics used for identifying the taxonomy of bivalve pediveliger larvae were the shell shape and size, presence of eye, and also the size and shape of the shell umbo.

### 2.2. Fixation of Larvae

For immunocytochemical staining, the samples were fixed in a 4% paraformaldehyde solution (PFA, pH 7.5) stabilized with methanol (ROTI^®^Histofix, Carl Roth Gmbh & Co. Kg, Karlsruhe, Germany) for 3 h at room temperature, and then the supernatant was removed, and the samples were washed with a 0.1% Triton X-100 solution (Sigma-Aldrich, St. Louis, MO, USA) in phosphate-buffered saline (PBS) (Sigma-Aldrich, St. Louis, MO, USA) three times for 10 min. After that, the excess supernatant was removed, and the washed material was dehydrated through a series of methanol solutions (Solveco, Rosersberg, Sweden) in PBS with increasing alcohol concentrations (25, 50, 75, 100% methanol and then again in 100% methanol, 10 min per each). The fixed material was stored in 100% methanol at a temperature of −25 °C.

### 2.3. Immunocytochemical Staining

The method of indirect immunocytochemistry adapted for our previous studies [9,31] was used for antibody staining of larvae. To prepare larvae for indirect immunocytochemical staining, they were transferred from 100% methanol to PBS through a series of PBS/methanol solutions with increasing PBS concentration. Then, the material was incubated for 1 h in a 5% ethylenediamide tetra-acetic acid (EDTA) solution (Sigma-Aldrich, St. Louis, MO, USA) at room temperature for decalcification of shells, washed with a PBS solution with 0.1% Triton X-100, and then, to reduce nonspecific antibody binding sites, incubated overnight at 4 °C in a blocking solution (1 × PBS, 10% normal donkey serum (NDS) (Jackson ImmunoResearch, Cambridge House, St. Thomas Place, UK), 1% bovine serum albumin (BSA) (Sigma, St. Louis, MO, USA), 1% Triton X-100, and 0.03% NaN_3_). Afterwards, larvae of all noted species (about 100 specimens per species) were incubated with primary antibodies in a blocking solution at 4 °C for 3 days. Then, the samples were washed in PBS with 0.1% Triton X-100 and incubated overnight at 4 °C with secondary antibodies in the blocking solution. The list of antibodies used in this study is provided in Table 1. These antibodies were also used in several other studies [9,12,32].

After incubation, the material was washed of secondary antibodies in the PBS solution with 0.1% Triton X-100 3–5 times for 10 min at room temperature. Then, the samples were dehydrated through a series of methanol solutions with increasing alcohol concentrations (50, 100% methanol, and then again 100% methanol, 5 min per each), the supernatant was removed, and a mixture of benzyl alcohol (BA) (Merck KGaA, Darmstadt, Germany) and benzyl benzoate (BB) (Sigma-Aldrich, St. Louis, MO, USA) at a ratio of 1:2 (BA/BB) was added immediately. Afterwards, the larvae were incubated for 30 min at room temperature for tissue discoloration. The material was distributed into glass-bottom plastic Petri dishes (MatTek Corporation, Ashland, MS, USA), one species per dish.

### 2.4. Antibodies

In this work, we used goat anti-serotonin antibodies combined with bovine serum albumin (BSA) and paraformaldehyde (PFA) (Immunostar, Hudson, WI, USA, #20079). According to the manufacturer’s statement, staining with this antiserum is eliminated by pretreatment with 25 μg of the same serotonin–BSA conjugate per 1 mL of diluted antibody. Control shows that pre-incubation of the antibody with the same conjugate (10 µg/mL, ImmunoStar, cat. no. 20081) at 4 °C overnight completely eliminates immunolabile serotonin staining in tissues. Pre-adsorption of the diluted antiserum with 10 mg/mL BSA overnight at 4 °C did not affect this staining, i.e., these antibodies recognized only serotonin but not BSA. Data from other studies have shown that these antibodies detect serotonin in adult bivalves and their planktotrophic larvae [7,8,12,14,33,34,35].

Rabbit antibodies to synthetic FMRFamide (Phe-Met-Arg-Phe-amide) conjugated with bovine thyroglobulin (Immunostar, #20091) were used to detect elements of the pediveliger nervous system. According to published data, the antibody reacts with FMRFamide of various animals including mollusks: gastropods, bivalves, and polyplacophora [12,14,33,36]. The manufacturer has confirmed that these antibodies react with antigens of chitons (*Mopalia muscosa*), gastropod mollusks (*Helix pomatia*, *Aplysia* sp., *Ilyanassa obsoleta*, *Lymnaea stagnalis*), and bivalves (*Mytilus trossulus*).

Mouse monoclonal antibodies to acetyl-alpha-tubulin (Santa Cruz Biotechnology, Santa Cruz, CA, USA, #sc-23950) were used to detect ciliary structures in pediveligers. According to the manufacturer, the antibodies are specific to acetyl-alpha-tubulin in mammalians, zebrafish, *Drosophila*, and *Xenopus*. The immunogen for the preparation of the primary antibody was isolated from acetylated alpha-tubulin of the axoneme of the sea urchin sperm flagellum. Antibodies to acetylated alpha-tubulin have been widely used to identify ciliated structures in larvae of annelids, mollusks, and nemertines [14,36,37]. These antibodies label the neuronal elements in some adult mollusks and annelids [12,38]. Moreover, larvae stained only with secondary antibodies were analyzed as a control for all primary antibodies.

### 2.5. Confocal Microscopy

The obtained preparations were examined under an LSM 780 confocal laser scanning microscope (Zeiss, Germany) equipped with a multiline laser (488, 555, and 647 nm), using the original Zen software (Black Edition) version 2.0. There were about 50 specimens per species analyzed, and then images of larvae (about 10 specimens per species) were taken in the Z-stack mode with an optical section thickness of 1 μm along the Z-axis and converted into projections in the maximum intensity mode. The obtained images were processed in the software Imaris 7.0 (Karolinska Institute, Stockholm, Sweden) and Photoshop 22.1.1 (USA).

## 3. Results

### 3.1. Species and Criteria

The species of bivalves considered below represent four ecological groups (Figure 1):
Burrowing forms (Figure 1A,B):
-*Callista brevisiphonata* (Carpenter, 1864), Japanese callista clam (Figure 1A);-*Mactromeris polynyma* (Stimpson, 1860), Arctic surf clam (Figure 1B).
Cemented forms (Figure 1C,D):
-*Pododesmus macrochisma* (Deshayes, 1839), green falsejingle (Figure 1C);-*Crassostrea gigas* (Thunberg; 1793); Pacific oyster (Figure 1D).
Byssally attached forms (Figure 1E,F):
-*Crenomytilus grayanus* (Dunker, 1853), Gray’s mussel (Figure 1E);-*Kellia japonica* (Pilsbry, 1895), Japanese kellia (Figure 1E).
Mobile forms (Figure 1G,H):
-*Mizuhopecten yessoensis* (Jay, 1857), Yesso scallop (Figure 1G);-*Azumapecten farreri* (Jones et Preston, 1904), Farrer’s scallop (Figure 1H).


The choice of these species was based on the characteristic features of their biology and the mode of life they lead in the adult stage. Serotonin and FMRFamide were selected as the main neurotransmitters for labeling, since these are classical markers used to detect the neuromorphology of several invertebrate taxa [36]. The nomenclature of neuronal structures described in the present study was used in accordance with the terms suggested in the international neuroanatomical glossary [39]. The neuromorphology and abbreviations of the neurostructures of larval bivalves were used based on accepted and published studies on the development of bivalve species [9,16,35].

### 3.2. Burrowing Forms of Bivalve Mollusks

#### 3.2.1. Larval Neuromorphology of *Callista brevisiphonata*

In the central nervous system (CNS) of Japanese callista clam (*Callista brevisiphonata* (Carpenter, 1864) (Figure 2A–C), the serotonin (5-HT)-immunopositive structures are the apical organ (AO) and the commissures of the CG and PG (Figure 2B,C). In the case of the AO, the stained elements are an aggregation of 5-HT-immunopositive cells and an aggregation of neurites extending from them and passing into paired bundles of the commissure of the CG. In the PG, paired serotonin-positive cells fan out from the center and connect in pairs via commissures, forming a ladder-shaped structure (Figure 2B).

Positive immunoreactivity to FMRFamide is observed in all ganglia (Figure 2D–F). The paired CG are connected via short, thin connectives to the paired PlG (Figure 2E). Large aggregations of FMRFamide-immunopositive cells connecting to each other via thick commissures and tightly concentrating at the center under the AO are distinguishable in the CG. Neurites extend from this center, forming a neuropil in the area of 5-HT-positive cells of the AO (Appendix A). FMRFamide-immunopositive PlG is connected to the PG and VG via the immunoassayed paired pleuro-pedal and pleuro-visceral connectives, respectively (Figure 2F). The FMRFamide-immunopositive PG is a pair of club-shaped clusters of cells connected via ladder-arranged commissures (Figure 2E). The VG, located in the posterior part of the body, is formed by paired aggregations of cells connected via thick arcuate commissures (Figure 2D,E). Double staining to FMRFamide and 5-HT show AO, CG, and PG (Appendix A).

The peripheral nervous system (PNS), stained for serotonin by antibodies, is represented by a stomatogastric neuron (sgn) and pedal nerves (pn) (Figure 2A,D). The sgn is localized in the stomach wall and is connected to the serotonin cells of the AO via a long neurite. We did not find an ens in callista clam’s pediveliger. Weak serotonin immunostaining was detected below the PG in the foot region, which can be interpreted as traces of pn (Figure 2A,C).

FMRFamide-immunopositive elements of the PNS are represented by dorsal (dn), pallial (pln), and velum (vn) neurons and by FMRFamide-positive nets of neurites in the region of anterior (aan) and posterior (pan) adductors and vn (Figure 2D-F). In the sgn, FMRFamide, unlike serotonin, is detected only in the neuron body and in the part of the neurite closest to the PlG connecting the nerve cell to the neuropil of the AO (Appendix A). The enteric nervous system (ens) was not detected. A relatively large FMRFamide-immunopositive aan and a pair of dn nearby are associated with PlG (Figure 2D). This aggregation of neurites originates from a pair of lateral branches of vn running along the velum margin up to the level of the paired clusters of the PlG cells. In the posterior part of the body, there are FMRFamide-immunopositive pln connected with the VG located nearby (Figure 2F).

#### 3.2.2. Larval Neuromorphology of *Mactromeris polynyma*

In surf clam (*Mactromeris polynyma* (Stimpson, 1860), the 5-HT-immunopositive parts of the CNS are the AO and the CG (Figure 3A–C). In the AO, the stained elements are a wide cluster of cells with paired neurites extending from its sides and uniting into a ring in the region of the CG commissure (Figure 3B,C).

In the surf clam’s CNS, antibodies against FMRFamide stain all the ganglia in the larva (Figure 3D–F). The group of CG cells has neurite processes on the sides, connecting to paired dense aggregations of neurons of the PlG (Figure 3E,F). The CG are also connected to each other via thick commissures (Figure 3F). Above the CG, there is an unpaired cluster of the AO neurites connecting to the CG via paired short connectives (Figure 3E,F). The nerve processes running downwards from the dense lateral clusters of the PlG cells are combined into neuropils of other paired lateral groups of neurons. The PlG are connected to the PG via pleuro-pedal connectives (Figure 3E,F). The PG is represented by paired aggregations of cells connected via two commissures anteriorly and posteriorly (Figure 3E). The VG communicates with the PlG via a pair of pleuro-visceral connectives, including groups of large, weakly stained FMRFamide cells localized on them (Figure 3E). The ganglion is a paired aggregation of cells connected via a narrow arcuate visceral commissure (Figure 3E).

The PNS structure in *M. polynyma* also shows a positive immunoreaction to serotonin (Figure 3A–C). A pair of neurites extends from the 5-HT-immunopositive cluster of cells of the AO to the anterior margin and merges in the region of the anterior adductor, forming a nerve net (aan) (Figure 3C). The single large sgn in the stomach wall is also associated with the AO (Figure 3C).

Staining with antibodies against FMRFamide revealed some elements of the PNS in *M. polynyma* (Figure 3D–F). Paired neurites extend from below the AO and form aan and also a pair of dn located nearby (Figure 3E). The paired vn run from the lower part of the PlG, along the anterior margin of the velum, to the dn. The body of the sgn can also be found in the stomach wall (Figure 3F). The neurites forming the foot innervation extend from the anterior horns of the PG (Figure 3F). The pan located near the ganglion is associated with the VG (Figure 3F). Below the adductor, there is a pair of small pallial nerve cells whose processes are oriented along the posterior margin of the velum (Figure 3F). At the velum margin under the pn, small cells with nerve processes oriented towards the posterior margin of the body are discernible (Figure 3D). Above the VG, there is a long FMRFamide-immunopositive ens extending to the region of the body near the umbo (Figure 3F). At the end of its trajectory, there is a small FMRFamide-positive nerve cell at the level of aggregation of large cells of the VG.

### 3.3. Cemented Forms of Bivalve Mollusks

#### 3.3.1. Larval Neuromorphology of *Pododesmus macrochisma* (Deshayes, 1839), Green Falsejingle

In green falsejingle, such CNS elements as the AO and the PG show a 5-HT-immunopositive reaction (Figure 4A–C). In the AO, an aggregation of several large cells is stained, with one flask-shaped cell located at some distance from the rest of the aggregation (Figure 4C). A pair of diffusely stained neurites extend from the AO to the CG (Figure 4B). In the PG, paired groups of small cells can be distinguished (Figure 4B,C). These are connected to their counterparts via commissures, eventually forming a ladder-like structure of PG (Figure 4B).

The CNS elements in *P. macrochisma* larvae show immunopositive FMRFamide staining (Figure 4D–F). In the AO, an aggregation of cells and their neuropils is detected in the same region as 5-HT-immunopositive cells (Figure 4E,F). Below it, there is a closely located merging pair of CG connected via a thick, short commissure (Figure 4E). These communicate through paired thick bundles of neurites with the paired smaller aggregations of PlG cells. Several small FMRFamide-positive aggregations are located among these bundles (Figure 4E). The PlG and PG communicate via paired diffuse pleuro-pedal connectives (Figure 4F). The PG is represented by paired large butterfly-shaped aggregations (Figure 4E). The VG communicates with the PlG through paired long and thick pleuro-visceral connectives (Figure 4F). VG is represented by a long arcuate bundle of neurites, which is a visceral commissure connecting small paired groups of cells (Figure 4E,F).

The 5-HT-immunopositive PNS elements of green falsejingle larvae can be detected (Figure 4A–C). The aan extends from the serotonin aggregation of cells of the AO to the anterior margin of the body (Figure 4C). A diffuse aggregation of traces of neurites can be detected in the lower part of the body. Close to the VG’s location are pan from which nerves with cells extend towards the umbo, which is observed near the posterior margin (Figure 4A,B). 5-HT immunoreactivity is present in the paired pedal ganglion, the cells of which form a ladder-like structure (Figure 4B,C).

The immunoreactivity to FMRFamide is detected to a much greater extent in the neurostructures related to the PNS (Figure 4D–F). The aan extends from the AO towards the anterior part of the umbo (Figure 4D). The dn, from which the thin vn run along the anterior margin of the velum, are near the adductor (Figure 4D,E). Moreover, two small nerve cells of the velum are found below the PlG, near the vn (Figure 4D). A paired net of pn is localized under the PG (Figure 4F). It also includes the pan and a pair of pln located nearby (Figure 4F). FMRFamide immunoreactivity was not detected in sgn.

#### 3.3.2. Larval Neuromorphology of *Crassostrea gigas* (Thunberg, 1793), Pacific Oyster

In Pacific oyster late larvae, the AO, PlG, and, possibly, AcG show positive immunoreaction to serotonin (Figure 5A–C). In the AO, a large aggregation of cells and lateral paired vn extending from it are stained (Figure 5B). Paired unfused aggregations of 5-HT-immunopositive PlG cells are found on the lateral nerve cords (Figure 5B,C). A thin filament of neurite and paired lateral AcG cells are distinguished in the visceral commissure region (Figure 5B,C).

All ganglia of larvae also show immunopositive reaction to FMRFamide (Figure 5D–F). In the anterior part of the body, there is an unpaired large cluster of cells where, according to the ganglion structure, a “cap” can be distinguished, which is the AO, and a denser aggregation, the CG (Figure 5E,F). The latter communicates with the pairs of the PlG through long connectives (Figure 5E,F). The PG is paired and is formed by a relatively small unfused aggregation of cells communicating via thin pleuro-pedal connectives with the PlG (Figure 5E). In the VG, paired groups of cells connected via a large arcuate visceral commissure can be distinguished (Figure 5E,F).

In the oyster PNS, antibodies against serotonin stain a large and developed network of velum neurites and an aggregation of neurites in the anterior adductor region associated with the AO (Figure 5A–C).

The network of the vn associated with the AO shows an immunopositive reaction to FMRFamide (Figure 5D–F). The aan is associated with the AO, but part of it also originates from the cerebro-pleural connective (Figure 5D,E). A pair of pan with nerve cells located near the posterior margin of the body extend from the VG (Figure 5E,F). No PNS elements related to the digestive system such as sgn and ens were found in the oyster’s pediveliger.

### 3.4. Byssally Attached Forms of Bivalve Mollusks

#### 3.4.1. Larval Neuromorphology of *Crenomytilus grayanus* (Dunker, 1853), Gray’s Mussel

In Gray’s mussel, the AO and the PG of the CNS show a positive immunoreaction to serotonin (Figure 6A–C). The cluster of large cells of the AO, with paired neurites extending from its sides and continuing into the pleuro-pedal and pleuro-visceral connectives, stands out to the greatest extent (Figure 6B,C). The PG anlagen is represented by a small, weakly stained paired aggregation of cells (Figure 6C). No serotonin immunoreactivity was detected in CG.

FMRFamide-immunopositive elements of the CNS are the AO and the CG, PlG, PG, and VG (Figure 6D–F). The paired CG are connected via a large arcuate bundle of neurites with the central unpaired aggregation of the AO (Figure 6E). The CG is connected to the paired groups of PlG cells via paired lateral cerebro-pleural connectives (Figure 6E). The paired pleuro-visceral connectives, connecting them to the much narrower VG, extend from these groups of cells (Figure 6E). On the sides of the VG, there are small cells connected via a visceral thick commissure (Figure 6E). The PG is connected to the PlG via pleuro-pedal connectives (Figure 6F). It is represented by a paired group of cells connected via several pedal commissures, which gives the ganglion the shape of a butterfly (Figure 6E).

The mussel’s pediveliger PNS stained for serotonin is represented by branched vn and aan associated with the AO and also by a nerve branch towards the lower margin (Figure 6A–C). In the PG region, there are small fragmentary filaments of nerve processes that, most likely, can be interpreted as pn (Figure 6C). A pair of neurite networks of pan is located in the posterior part of the body (Figure 6A,B).

In mussel’s pediveliger, such PNS elements as the aan and vn extending from them along the velum anterior margin, associated with the AO, exhibit immunopositive reaction to FMRFamide (Figure 6D–F). A pair of neurite networks belonging to the pan are located in the immediate vicinity of the VG (Figure 6E,F). We did not detect sgn and ens as well as pln and pn in the PNS of the mussel’s pediveliger.

#### 3.4.2. Larval Neuromorphology of *Kellia japonica* (Pilsbry, 1895), Japanese Kellia

In Japanese kellia, the 5-HT-immunopositive elements of the CNS are the AO and the PG (Figure 7A–C). The AO is represented by a cluster of apically oriented cells, with dense bundles of neurites extending from its sides, and by a thin commissure communicating with the paired PlG (Figure 7B,C). In the PG, a group of small dorso-ventrally oriented cells was stained for serotonin (Figure 7B,C).

The elements of the CNS stained for FMRFamide are the AO and the CG, PlG, PG, and VG (Figure 7D–F). The AO consists of an unpaired aggregation of neurons and their processes, with the CG lying below (Figure 7E). CG is represented by a paired, almost fused cluster of neurons with paired dense bundles of neurites extending from it downwards and backwards to the PlG (Figure 7E,F). PlG is represented by paired lateral oblong aggregations of nerve cells not connected by any commissures (Figure 7E). Then, the PlG communicates with the VG through paired pleura-visceral connectives (Figure 7E,F). The VG is represented by paired aggregations of small cells connected to each other via short visceral commissures (Figure 7E,F). Then, the weakly stained small PG consisting of small cells of the same morphology follows, as described above for the 5-HT-immunopositive reaction (Figure 7F).

Some of the PNS elements show positive immunoreactivity to serotonin (Figure 7A–C). Paired neurites run from the AO towards the umbo (Figure 7C). The sgn, which is most likely related to the AO, can also be observed in the stomach wall (Figure 7A).

Immunostaining for FMRFamide is observed in a much larger part of the PNS (Figure 7D–F). At the anterior margin of the body, there is the aan associated with the AO and also the dn that send the vn along the margins of the body (Figure 7D). The body of an sgn is also observed in the stomach wall (Figure 7D). Paired networks of pan are located near the VG, and several nerve cells are slightly above: two in the immediate vicinity of the adductor and another two near the stomach (Figure 7D,E). The vn run below the VG, along the posterior margin of the velum (Figure 7F). No pn and pln were detected in *K. japonica* pediveliger.

### 3.5. Mobile Forms of Bivalves

#### 3.5.1. Larval Neuromorphology of *Mizuhopecten yessoensis* (Jay, 1857), Yesso Scallop

The AO and the PG and VG in Yesso scallop show immunoreactivity to serotonin (Figure 8A–C). In the AO, a cluster of cells is distinguished that extends bundles of paired lateral neurites to the paired PlG (Figure 8B). In the PG, the paired lateral groups of cells connecting to each other via ladder-arranged commissures contain serotonin (Figure 8B). The staining also revealed 5-HT immunoreactivity in thin neurites running through the VG’s commissure (Figure 8B,C).

The AO and the CG, PlG, PG, and VG showed FMRFamide-immunopositive reaction (Figure 8D–F). In the AO, an aggregation of FMRFamide is observed around the above-described 5-HT-immunopositive cells, with a dense commissure of paired CG located tightly near it (Figure 8E). The paired broad lateral bundles of neurites extend from them to the aggregation of neurons of the PlG. These aggregations are interconnected via a thin commissure (Figure 8E). Double staining for FMRFamide and 5-HT shows AO, CG, PG, and VG (Appendix A)

The FMRFamide-immunopositive anterior horns of the PG are connected to paired aggregations of PlG neurons via paired thin pleuro-pedal connectives (Figure 8E,F). The PG is formed by a butterfly-shaped aggregation of cells, with the anterior horns (that extend pleuro-pedal connectives) facing the apical pole and the posterior horns (that extend neurites to the foot region) directed to the VG (Figure 8E). The PlG communicates with the VG through paired pleuro-visceral connectives (Figure 8E,F). The VG represents paired lateral groups of small cells connected via a thick arcuate commissure (Figure 8E).

The 5-HT-immunopositive elements of the PNS of Yesso scallop are the apical neurites extending from the respective side of the serotonin cells of the AO and the pn in the foot region associated with the PG (Figure 8B,C).

In *M. yessoensis*, the immunopositive staining for FMRFamide was found in a much greater number of morphological structures (Figure 8D–F). The stained elements of the PNS associated with the AO are the aan and the dn that send vn along the anterior margin of the velum (Figure 8E,F). The neurites forming the ens in the digestive system region also extend from the AO (Figure 8E). The sgn is undetectable by immunoreaction for FMRFamide and serotonin. The FMRFamide-immunopositive pn branch from the PG ventrally into the foot region (Figure 8F). The paired lateral networks of the pan are associated with the VG: these originate from the lateral aggregations of ganglion cells and are located nearby, at the posterior margin of the body (Figure 8F). Small bundles extend dorsally from these neurites into the ens region (Figure 8F). Below the VG, there is a quite concentrated aggregation of neurites near the pn (Figure 8F). We did not detect any immunoreactivity in pln.

#### 3.5.2. Larval Neuromorphology of *Azumapecten farreri* (Jones et Preston, 1904), Farrer’s Scallop

The Farrer’s scallop shows a positive immunoreaction to serotonin in the structures that constitute the CNS (Figure 9A–C).

In the AO, a large group of cells and paired lateral bundles of neurites running to the CG are stained (Figure 9B,C). The CG and PG communicate through paired cerebro-pedal connectives (Figure 9B,C). The PG also represents a somewhat loose group of cells having a butterfly-like shape (Figure 9B). The CG is also connected to the VG via paired cerebro-visceral connectives (Figure 9B,C). An aggregation of neurite bundles is found in the visceral commissure region (Figure 9B,C).

In Farrer’s scallop, all FMRFamide-immunopositive ganglia of the CNS are clearly identified (Figure 9D–F). The AO tightly adjoins the almost-fused CG (Figure 9E,F). The CG send paired lateral connectives communicating with another pair of cell groups that belong to the PlG (Figure 9E,F). The PlG is connected to the PG via a pair of pleuro-pedal connectives (Figure 9E,F). The small PG consists of paired clusters of small cells connecting to each other via a short pedal commissure (Figure 9E). Moreover, the PlG communicates with the VG through paired pleuro-visceral connectives (Figure 9E,F). A paired aggregation of small cells, connected via a flat, diffuse visceral commissure, can be distinguished in the VG (Figure 9E,F).

The PNS in Farrer’s scallop is positively stained with antibodies against serotonin and is represented by various structures (Figure 9A–C). Paired networks of neurites, with vn branching off them and running all along the anterior margin of the body, extend from the AO to the anterior margin (Figure 9A). 5-HT-immunopositive diffuse pn run downwards from the PG (Figure 9C). The network of neurites located below the VG, from which branches of pallial nerves extend all along the posterior margin of the velum, is associated with the VG (Figure 9C).

Antibodies against FMRFamide detect not as many PNS structures as those against serotonin (Figure 9D–F). A paired aggregation of neurites of the aan, presumably associated with the AO, is revealed at the anterior margin of the larva’s body (Figure 9D). Weak traces, probably belonging to the ens, are observed in the stomach region (Figure 9F). A pair of networks of the posterior adductor neurites is found near the VG (Figure 9D). No sgn was detected, the same as pln.

## 4. Discussion

The bivalve CNS has bilaterally symmetrical ganglionic organization. A vast number of neurotransmitters of different origins and performing various functions have been recorded from the nervous system of adult bivalves. Cholinergic neurons are found in all ganglia of adult mollusks [40]. An increase in their number in one or another ganglion indicates the degree of activity of the body part that this ganglion innervates: in burrowing forms, the greatest number of cholinergic neurons is located in the PG; in swimming forms, in the VG [20,21]. The functions of acetylcholine are heartbeat regulation [41,42], the catch state of adductors [12,43], and also ciliary beating [44]. Acetylcholine has also been found to be involved in the nervous system’s development in larval *M. galloprovincialis* [45] and *C. gigas* [9].

Monoamines such as serotonin, dopamine, tryptamine, norepinephrine, epinephrine, and histamine are also recorded from all ganglia of adult mollusks. Their concentration varies between different parts of the CNS at the interspecific level [20]. It has been reported that in Gray’s mussel (*C. grayanus*), serotonin is present in the cerebro-PlG and PG, while tyrosine hydroxylase, an enzyme that catalyzes the dopamine synthesis, is present in all ganglia, as well as norepinephrine [12]. Histamine-positive immunoreactivity has been detected in all ganglia of Baltic clam (*Macoma balthica*) [46].

In the bivalve CNS, neurons are found that contain γ-aminobutyric acid (GABA). Their presence in all ganglia of *C. grayanus* and *M. balthica* has been confirmed, with the largest number of neurons recorded from the VG and PG, respectively [12,47]. In *M. balthica*, octopamine-positive immunoreactivity is detected in all ganglia. Greater numbers of octopamine-stained neurons, grouped into clusters, are found in the PG than in the rest of the ganglia [47].

In larvae, the main neurotransmitters are serotonin, FMRFamide, and catecholamines such as dopamine, epinephrine, and norepinephrine, as well as a small amount of cardiac peptide-B. Catecholamines, serotonin, and FMRFamide are present in cells associated with velum, thus playing a role in the functions of swimming and feeding [10,48,49]. Serotonin-positive axons are usually observed only in cells of the AO, while catecholamines and, in particular, the neuropeptide FMRFamide can occur in many neuronal structures of the larval body [7,10].

Catecholamines such as epinephrine and norepinephrine are reported to have a potential effect on the metamorphosis of bivalve larvae. It has been found that the induction of metamorphosis in *C. gigas* (Pacific oyster) larvae exposed to epinephrine occurs under the regulation of α-adrenergic receptors [50]. In veligers of *Meretrix meretrix* (hard clam), β-adrenergic neurons are found in the apical ganglion that, presumably, may play a crucial role in metamorphosis regulation during larval settlement [51]. Besides the CNS, catecholamines have also been detected in the PNS of pediveligers: in neurons of the velum, foot, in the mouth region, and in the forming gills of settled larvae [52]. Thus, these neurotransmitters are assumed to play a key role in the regulation of swimming, movement on the substrate, and feeding.

To date, a number of studies have been published that describe the larval nervous systems of species belonging to the ecological groups of bivalves considered in the present report: the burrowing *Mya arenaria* [53] and *Spisula soldissima* [54], the cemented *Crassostrea virginica* [16] and *C. gigas* [9], the byssally attached *Mytilus trossulus* [7] and *M. edulis*, and also the swimming *Placopecten magellanicus* [52] and *Azumapecten farreri* [10]. Nevertheless, in all studies, there is no clear correlation between ecological groups and larval neurostructures. Most likely, the reason is that bivalves begin to develop their species-specific ecological niche from the juvenile stage of development, while at the larval stage they lead a pelagic lifestyle that differs little among species.

Unfortunately, there are not so many comparative studies on the neurobiology of bivalves. Kotsyuba’s paper on the study of adult bivalves’ response to hypoxia and elevated temperature showed that adult females of *C. grayanus* (byssally attached), compared to other species (*M. yessoensis* and *Peronidia venulosa*) have a much higher number of cells containing nitric oxide (NO), a marker of stress response in neurostructures [55,56]. At the same time, *P. venulosa* (burrowing) has rather small numbers of NO-positive cells only after exposure to stress factors, while in the swimming *M. yessoensis* NO-positive neurons were not detected either in the control or after stress exposure, yet it exhibits a motor response to stress (Kotsuyba, 2008). Presumably, such variation in NO content in ganglia in these species may be due to their lifestyle and ability to adapt to stressful environmental conditions: *C. grayanus*, being attached with a byssus, is confined to one location for almost its entire life, and as a response to increased temperature or hypoxia it is left to them to physiologically adapt to stress; meanwhile, *P. venulosa* and *M. yessoensis* have much higher mobility, especially *M. yessoensis*, which allows them to physically move to more optimal conditions and avoid stress. However, due to the small number of similar studies, it is too early to draw unambiguous conclusions, and further studies on other species from different ecological groups are needed to confirm this hypothesis.

The influence of a particular lifestyle can be reflected in the shell architecture of the adult mollusk. Each ecological group is characterized by specific shell shapes corresponding to the substrate on which the species lives [2,57].

Based on the above-cited publications and our data presented here, we can state that all of these species have retained the general tetraneural plan of the CNS structure with paired cerebral, PlG, PG, and VG.

In a study on Pacific oyster [9], the structure indicated as the “gill nerves” is here referred to as the “pedal ganglion”, since the true gills in bivalves appear at the pediveliger larva stage and are innervated by the VG nerves. As can be clearly seen here, the described structure has no communication with the VG. The PG of a similar localization is found in *C. virginica* [16].

In a study on the nervous system structure in pediveliger larvae of *Mya arenaria* [53], the authors attribute the dn described here to the true CG and the PlG to the AO. This is explained as follows: the dn, which send neurites towards the AO, are initially formed during the trochophore stage, while the cerebro-PlG is formed only at the following stages of development, and the AO and the associated vn are subsequently resorbed. We cannot agree with this statement, because it does not explain the tetraneural plan of the bivalve nervous system structure, according to which the CG in adults communicates with the PG and VG through the cerebro-pedal and cerebro-visceral connectives, respectively, as evidenced by our data presented here. It is also reported that larvae of a number of mollusks, e.g., of the class Gastropoda, are characterized by the localization of the AO above the cerebro-PlG, and only the part belonging to the AO disappears during embryogenesis [36].

The evolution of the pediveliger nervous system of the considered bivalve species is presented in Figure 10. The decoding of all nerve elements is presented in the following schematic picture (Figure 11).

We did not find any unambiguous correspondence between the structure of the nervous system and the ecological niche occupied by a particular species. However, the evolution of the innervation of the digestive system is noticeable, starting with the enteric nervous system in *A. farreri* and *M. yessoensis* (Pectinda) and ending with the stomatogastric neuron in *K. japonica*, *M. polynyma*, and *C. brevisiphonata* (Heterodonta). The absence of any enteric innervation in *C. gigas* (Osterida) and *C. grayanus* (Mytilida) may suggest its disappearance in these species during evolution.

The nerve cell in the stomach wall referred to as the “stomatogastric nerve cell”, besides the species considered in the present study (*C. brevisiphonata*, *M. polynyma*, and *K. japonica*), has also been found in a byssally attached freshwater bivalve, the zebra mussel *Dreissena polymorpha* [35]. The authors suggest that this cell may be somewhat responsible for regulating the processes related to the stomach, justifying their opinion by the location of the neuron being tightly close to the stomach. The authors cite also the papers where they claim that a similar structure, referred to as an “abdominal ganglion”, was found in the byssally attached *M. edulis* as a result of staining for catecholamine [52]. Here, the above-described structure is not related to the sgn, since it communicates with the PG and has nerve processes innervating the gills, which is more characteristic of the VG. In a study on *M. trossulus* [7], a structure similar to the sgn was stained for catecholamine.

We have identified a number of PNS structures whose presence varied significantly between different groups (Figure 10). The serotonin/FMRFamide-immunopositive sgn, associated with the apical ganglion located in the stomach wall region, was found in callista clam, surf clam, green falsejingle, and kellia. In a paper that mentions a similar structure [35], staining was only serotonin-positive, whereas in our case the nerve cell body showed FMRFamide immunoreactivity. Its function remains unknown. Green falsejingle has two nerve cells in the stomach region. Most likely, there is a certain relationship between this structure and the enteric nerve net in Yesso and Farrer’s scallops, since these three species belong to a single order, the Pectinida.

Serotonin-immunopositive accessory ganglia have been found in oysters, as well as the innervation in the VG region in scallops (Figure 10). It was previously stated that accessory ganglia provide eye innervation in adult bivalves including scallops [19]. Accessory ganglia have been found near the VG in *A. farreri* [10]. Among the species that we studied, accessory ganglia or serotonin-positive innervation are present in larvae with eyes, i.e., in Yesso and Farrer’s scallops and oyster. It is likely that accessory ganglia are associated with visual perception not only in adults but also in the larval stage of bivalves.

As has been reported earlier, in the velum of green falsejingle pediveligers (*P. macrochisma*), several nerve cells are detected that show positive immunoreaction to FMRFamide, which is not observed in pediveligers of the other species considered here. Bivalve larvae are characterized by the presence of a number of velum nerve cells associated with the apical ganglion that are stained for serotonin and catecholamines [52,53].

## 5. Conclusions

The external and internal bivalve larval structures may indicate adaptation to different conditions of their habitat; therefore, study of the larval systems of organs will provide essential data for further monitoring the plankton fauna of seas. Here, both morphologically identical structures and minor distinctive (specific) features of larval neuromorphology have been found in all larvae at the late developmental stages. This can be explained by the fact that the ecology-related features in the nervous system structure begin to appear at later stages, during/after the larval settlement on the substrate where the main metamorphosis occurs. It is obvious that the neuromorphological features are associated with the adaptations emerging in pediveliger larvae and juveniles. Furthermore, the data obtained change the accepted views concerning the reduction of the AO at the pediveliger stage. Judging from the morphology of pediveligers of the eight species under study, the AO and cerebro-(pleural) ganglion are present at the late stages of larval development. Thus, the study of the development of the nervous system structure at the late stages is evidently a relevant issue and a promising subject of further research.

## Figures and Tables

**Figure 1 biology-12-01341-f001:**
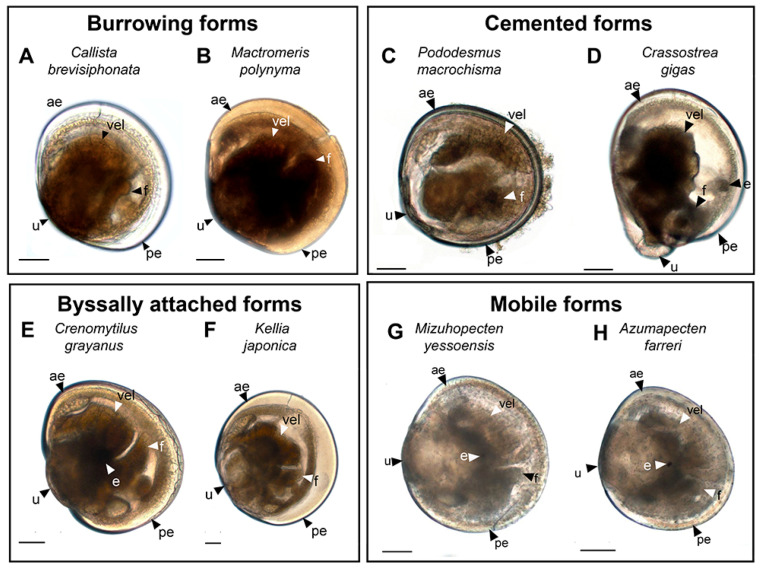
General view of pediveligers of burrowing adult bivalves (*Callista brevisiphonata*) (**A**) and *Mactromeris polynyma* (**B**); pediveligers of cemented adult bivalves (*Pododesmus macrochisma*) (**C**) and Pacific oyster (*Crassostrea gigas*) (**D**); pediveligers of byssally attached adult bivalves Gray’s mussel (*Crenomytilus grayanus*) (**E**) and Japanese kellia (*Kellia japonica*) (**F**); and pediveligers of mobile adult bivalves Yesso scallop (*Mizuhopecten yessoensis*) (**G**) and Farrer’s scallop (*Azumapecten farreri*) (**H**) under light microscopy. The letter designations are as follows: ae, anterior end; e, eye; f, foot; pe, posterior end; u, umbo; vel, velum. Scale bar = 50 µm.

**Figure 2 biology-12-01341-f002:**
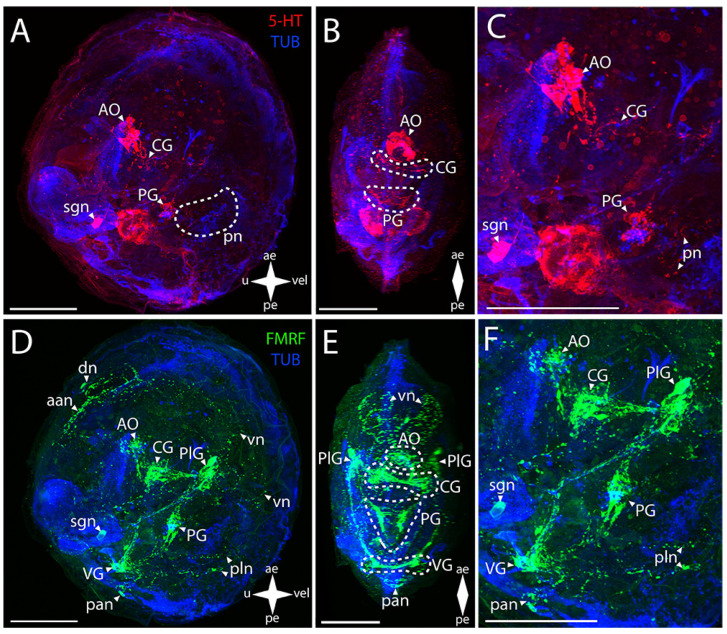
General structure of the nervous system in the Japanese callista clam’s pediveliger (*Callista brevisiphonata*). Red color indicates serotonin; green color, FMRFamide; blue color, tubulin. (**A**–**C**) Immunoreactive staining for serotonin; (**D**–**F**) Immunoreactive staining for FMRFamide; (**A**,**C**) lateral view; (**B**,**D**) umbo view; (**C**,**F**) magnifications of the CNS of the pediveliger. The letter designations are as follows: AO, apical organ; aan, anterior adductor nerves; CG, cerebral ganglion; dn, dorsal neurons; pan, posterior adductor nerves; PG, pedal ganglion, PlG, pleural ganglion; pln, pallial neurons; pn, pedal nerves; sgn, stomatogastric neuron; VG, visceral ganglion; vn, velum nerves. Crossed arrows indicate orientation of the pediveliger: ae, anterior end; pe, posterior end; u, umbo; vel, velum. Scale bar = 50 µm.

**Figure 3 biology-12-01341-f003:**
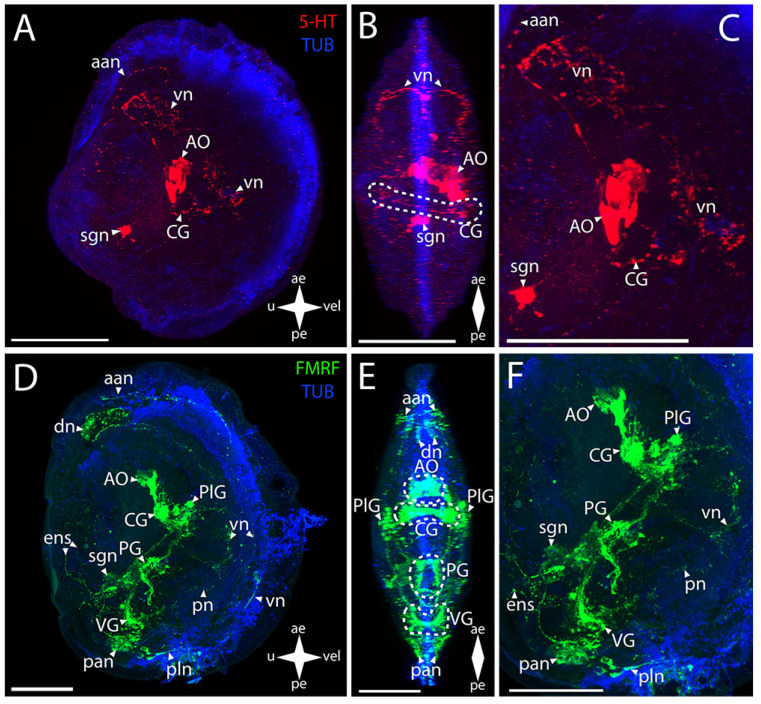
General structure of the nervous system in the Arctic surf clam’s pediveliger (*Mactromeris polynyma*). Red color indicates serotonin; green color, FMRFamide; blue color, tubulin. (**A**–**C**) Immunoreactive staining for serotonin; (**D**–**F**) Immunoreactive staining for FMRFamide; (**A**,**C**) lateral view; (**B**,**D**) umbo view; (**C**,**F**) magnifications of the CNS of the pediveliger. The letter designations are as follows: AO, apical organ; aan, anterior adductor nerves; CG, cerebral ganglion; dn, dorsal neurons; ens, enteric nervous system; pan, posterior adductor nerves; PG, pedal ganglion, PlG, pleural ganglion; pln, pallial nerves; pn, pedal nerves; sgn, stomatogastric neuron; VG, visceral ganglion; vn, velum nerves. Crossed arrows indicate orientation of the pediveliger: ae, anterior end; pe, posterior end; u, umbo; vel, velum. Scale bar = 50 µm.

**Figure 4 biology-12-01341-f004:**
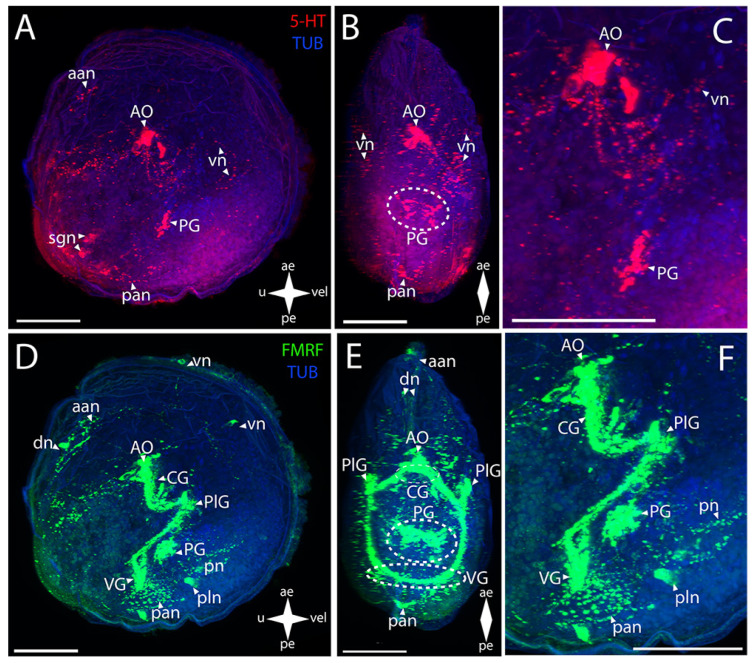
General structure of the nervous system in green falsejingle’s pediveliger (*Pododesmus macrochisma*). Red color indicates serotonin; green color, FMRFamide; blue color, tubulin. (**A**–**C**) Immunoreactive staining for serotonin; (**D**–**F**) Immunoreactive staining for FMRFamide; (**A**,**C**) lateral view; (**B**,**D**) umbo view; (**C**,**F**) magnifications of the CNS of the pediveliger. The letter designations are as follows: AO, apical organ; aan, anterior adductor nerves; CG, cerebral ganglion; dn, dorsal neurons; pan, posterior adductor nerves; PG, pedal ganglion, PlG, pleural ganglion; pln, pallial nerves; pn, pedal nerves; sgn, stomatogastric neuron; VG, visceral ganglion; vn, velum neurons. Crossed arrows indicate orientation of the pediveliger: ae, anterior end; pe, posterior end; u, umbo; vel, velum. Scale bar = 50 µm.

**Figure 5 biology-12-01341-f005:**
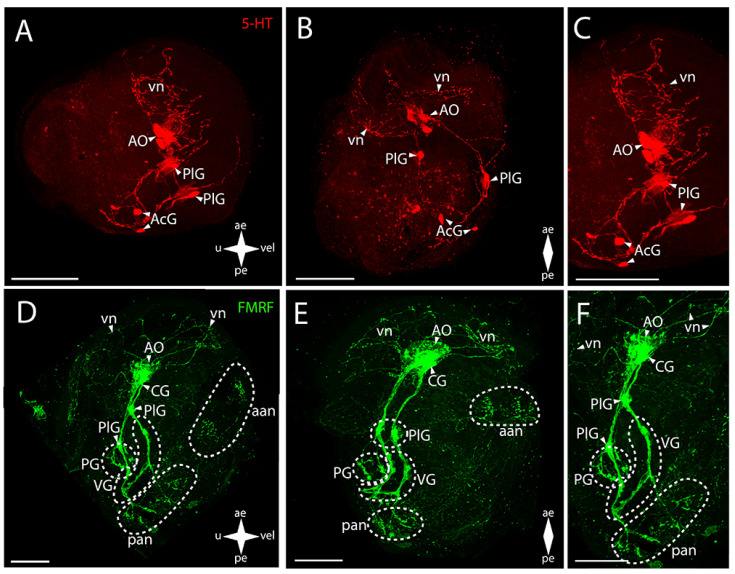
General structure of the nervous system in Pacific oyster’s pediveliger (*Crassostrea gigas*). Red color indicates serotonin; green color, FMRFamide. (**A**–**C**) Immunoreactive staining for serotonin; (**D**–**F**) Immunoreactive staining for FMRFamide; (**A**,**C**) lateral view; (**B**,**D**) umbo view; (**C**,**F**) magnifications of the CNS of the pediveliger. The letter designations are as follows: AO, apical organ; aan, anterior adductor nerves; CG, cerebral ganglion; pan, posterior adductor nerves; PG, pedal ganglion, PlG, pleural ganglion; VG, visceral ganglion; vn, velum nerves. Crossed arrows indicate orientation of the pediveliger: ae, anterior end; pe, posterior end; u, umbo; vel, velum. Scale bar = 50 µm.

**Figure 6 biology-12-01341-f006:**
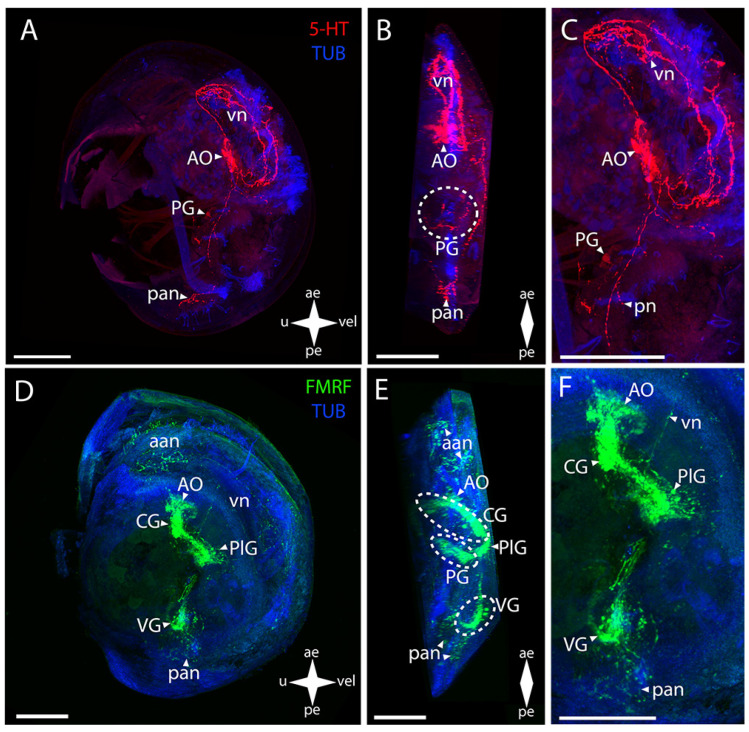
General structure of the nervous system in Gray’s mussel’s pediveliger (*Crenomytilus grayanus*). Red color indicates serotonin; green color, FMRFamide; blue color, tubulin. (**A**–**C**) Immunoreactive staining for serotonin; (**D**–**F**) immunoreactive staining for FMRFamide; (**A**,**C**) lateral view; (**B**,**D**) umbo view; (**C**,**F**) magnifications of the CNS of the pediveliger. The letter designations are as follows: AO, apical organ; aan, anterior adductor nerves; CG, cerebral ganglion; pan, posterior adductor nerves; PG, pedal ganglion, PlG, pleural ganglion; VG, visceral ganglion; vn, velum nerves. Crossed arrows indicate orientation of the pediveliger: ae, anterior end; pe, posterior end; u, umbo; vel, velum. Scale bar = 50 µm.

**Figure 7 biology-12-01341-f007:**
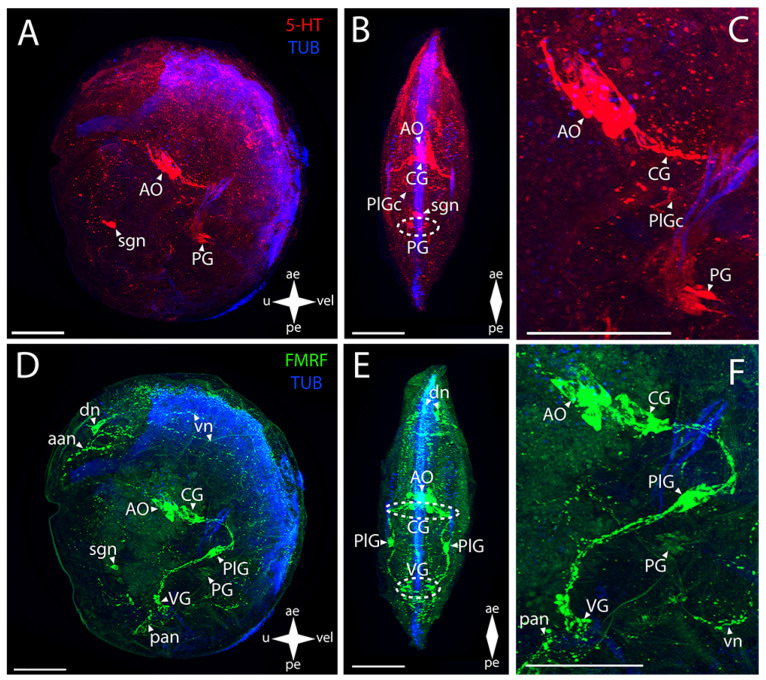
General structure of the nervous system in Japanese kellia (*Kellia japonica*). Red color indicates serotonin; green color, FMRFamide; blue color, tubulin. (**A**–**C**) Immunoreactive staining for serotonin; (**D**–**F**) immunoreactive staining for FMRFamide; (**A**,**C**) lateral view; (**B**,**D**) umbo view; (**C**,**F**) magnifications of the CNS of the pediveliger. The letter designations are as follows: AO, apical organ; aan, anterior adductor nerves; CG, cerebral ganglion; dn, dorsal neurons; VG, visceral ganglion; pan, posterior adductor nerves; PG, pedal ganglion; PlG, pleural ganglion; PlGc, pleural ganglia commissure; pn, pedal nerves; sgn, stomatogastric neuron; VG, visceral ganglion; vn, velum nerves. Crossed arrows indicate orientation of the pediveliger: ae, anterior end; pe, posterior end; u, umbo; vel, velum. Scale bar = 50 µm.

**Figure 8 biology-12-01341-f008:**
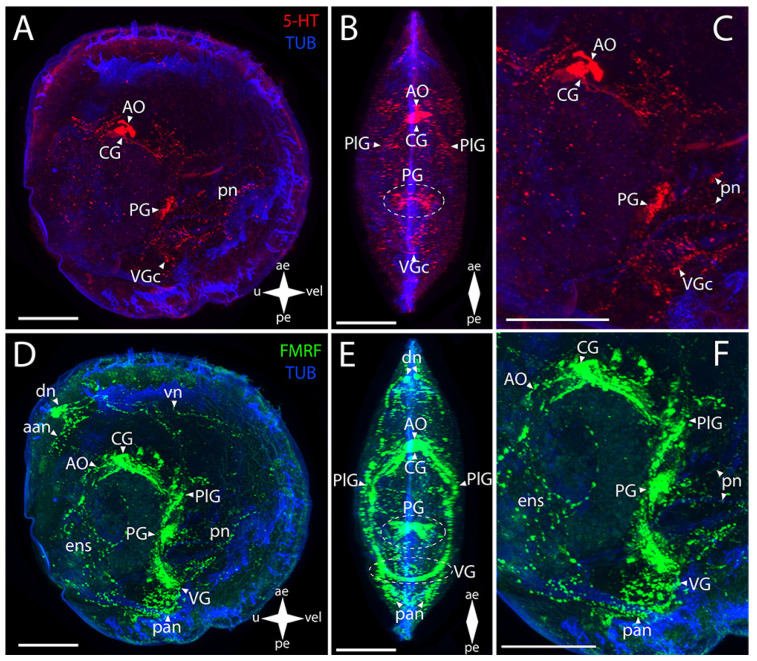
General structure of the nervous system in Yesso scallop’s pediveliger (*Mizuhopecten yessoensis*). Red color indicates serotonin; green color, FMRFamide; blue color, tubulin. (**A**–**C**) Immunoreactive staining for serotonin; (**D**–**F**) immunoreactive staining for FMRFamide; (**A**,**C**) lateral view; (**B**,**D**) umbo view; (**C**,**F**) magnifications of the CNS of the pediveliger. The letter designations are as follows: AO, apical organ; aan, anterior adductor nerves; CG, cerebral ganglion; dn, dorsal neurons; ens, enteric nervous system; VG, visceral ganglion; pan, posterior adductor nerves; PG, pedal ganglion; PlG, pleural ganglion; pn, pedal nerves; VG, visceral ganglion; vn, velum nerves. Crossed arrows indicate orientation of the pediveliger: ae, anterior end; pe, posterior end; u, umbo; vel, velum. Scale bar = 50 µm.

**Figure 9 biology-12-01341-f009:**
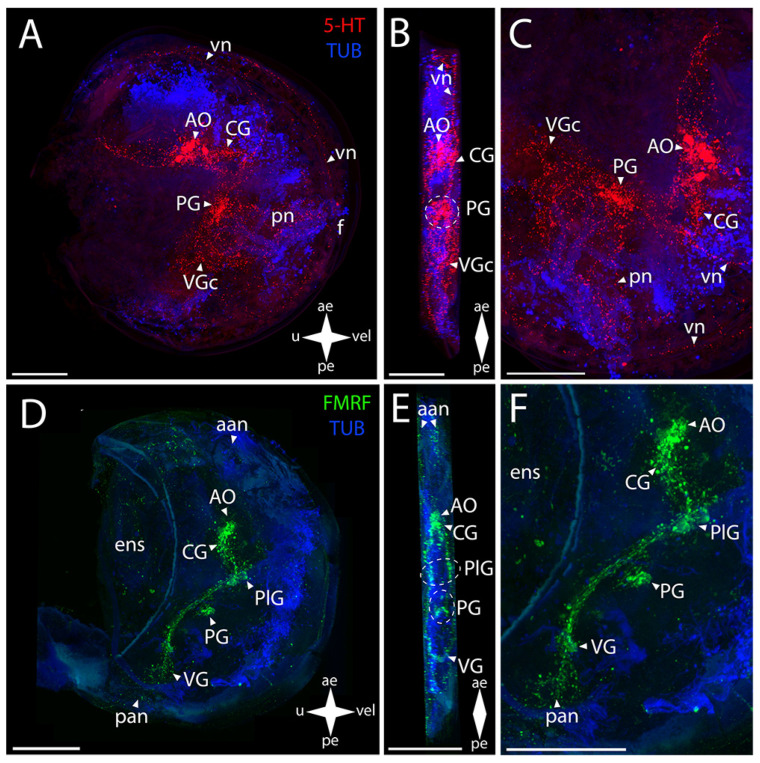
General structure of the nervous system in Farrer’s scallop’s pediveliger (*Azumapecten farreri*). Red color indicates serotonin; green color, FMRFamide; blue color, tubulin. (**A**–**C**) Immunoreactive staining for serotonin; (**D**–**F**) immunoreactive staining for FMRFamide; (**A**,**C**) lateral view; (**B**,**D**) umbo view; (**C**,**F**) magnifications of the CNS of the pediveliger. The letter designations are as follows: AO, apical organ; aan, anterior adductor nerves; CG, cerebral ganglion; ens, enteric nervous system; VG, visceral ganglion; pan, posterior adductor nerves; PG, pedal ganglion; PlG, pleural ganglion; pn, pedal nerves; VG, visceral ganglion; vn, velum nerves. Crossed arrows indicate orientation of the pediveliger: ae, anterior end; pe, posterior end; u, umbo; vel, velum. Scale bar = 50 µm.

**Figure 10 biology-12-01341-f010:**
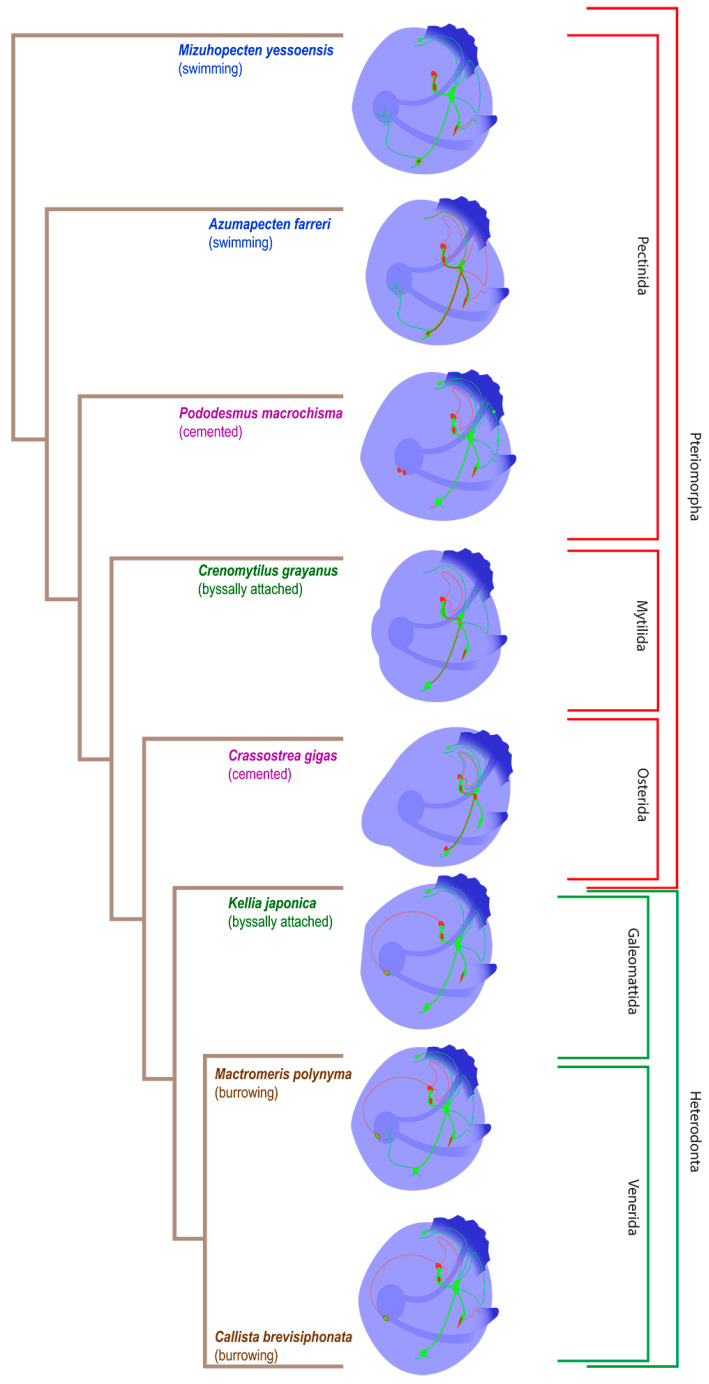
The phylogenetic tree of pediveliger’s nervous system of the bivalve species reviewed in this study, based on the data of [58]. Green is for FMRFamide, red is for 5-HT, and blue is for alpha-acetylated tubulin.

**Figure 11 biology-12-01341-f011:**
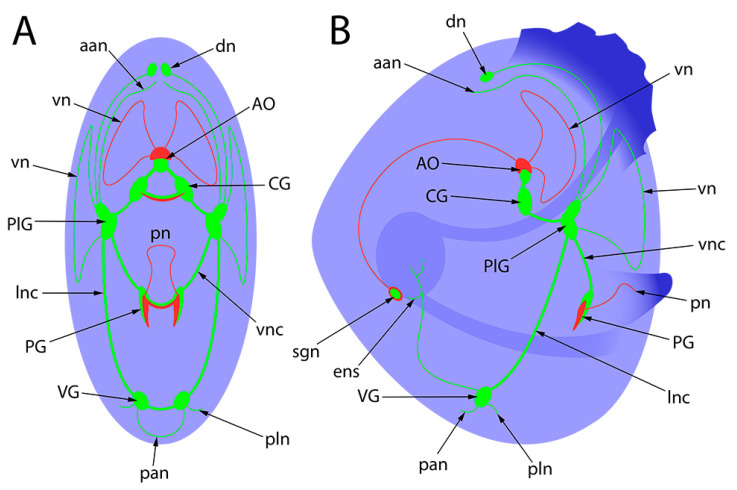
General schematic presentation of the nervous system of bivalves’ pediveligers. Green is for FMRFamide, red is for 5-HT, and blue is for alpha-acetylated tubulin. (**A**)—umbo view; (**B**)—lateral view. aan, anterior adductor nerve; AO, apical organ; CG, cerebral ganglia; dn, dorsal neurons; ens, enteric nervous system; lnc, lateral nerve cord; pan, posterior adductor nerve; PG, pedal ganglia; PlG, pleural ganglia; pln, pleural nerve; pn, pedal nerve; sgn, stomatogastric neuron; VG, visceral ganglia; vn, velum nerve; vnc, ventral nerve cord.

**Table 1 biology-12-01341-t001:** List of antibodies used.

Antibodies	Immunized Animal	Titer	Manufacturer, Country	No. of Antibody
**Primary antibodies**				
Antibodies against serotonin (5-hydroxytryptamine, 5-HT)	Goat	1:500–1:1000	Immunostar, Hudson, WI, USA	20079
Antibodies against FMRFamide (cardio-excitatory peptide)	Rabbit	1:500–1:1000	Immunostar, Hudson, WI, USA	20091
Antibodies against acetylated (α-tubulin)	Mouse	1:500–1:1000	Santa Cruz Biotechnology, Santa Cruz, CA, USA	sc-23950
**Secondary antibodies**				
Antibodies against goat blood (Alexa Fluor 555 donkey anti-goat IgG (H + L))	Donkey	1:500–1:1000	Invitrogen, Eugene, OR, USA	A21432
Antibodies against rabbit blood serum (Alexa Fluor 488 donkey anti-rabbit IgG (H + L))	Donkey	1:500–1:1000	Invitrogen, Eugene, OR, USA	A21206
Antibodies against mouse blood serum (Alexa Fluor Plus 647 donkey anti-mouse IgG)	Donkey	1:500–1:1000	Invitrogen, Eugene, OR, USA	A31571

## Data Availability

Not applicable.

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
