# Peer review of "Comparative Neuroanatomy of Pediveliger Larvae of Various Bivalves from the Sea of Japan"

_biology, 2023, doi:10.3390/biology12101341_

Round 1
Reviewer 1 Report
The paper by Nikishenko et al. represents an original descriptive and comparative study of the nervous system in different bivalve species at the pediveliger stage. The research is interesting with a clear and simple design. I really enjoyed reading it and would recommend for publication. However, I found some points that must be improved before publishing. Below you will find a list of my comments.
Legend for figure 1: abbreviation list is missing.
The figures themselves need to be improved. Sometimes it is not easy to find the structures, mentioned in the text. The abbreviations help a lot, but they are mostly about central nervous system and only few of them describe the PNS. Please, if it is not too hard, provide a bit more abbreviations helping to assess the details of the peripheral nervous system.
The lists of abbreviations in the legends are arranged in a non-alphabetic order. This makes them difficult to find.
Legend to Figure 5: What is ag – apical ganglion? Should it be the Apical organ? Please correct it here as well as on the other legends. Also, there is no “dn” and “pn” on the images. The same is for the legend for figure 6. It looks like a copy-paste artefact.
Line 253: The word “morphostructure” looks a bit confusing, as far as I know this term is referred rather to geology, than biology.
Line 288: The term neuropil is used a bit incorrectly. I’m afraid that it is impossible to apply the term neuropil to the group of neurites of the single cell. If my memory doesn’t let me down, the neuropil of the apical organ is called “apical neuropil”. Do you mean that the neurites of these cells contribute to it? To be honest, I think I understand what do you mean, but please, rephrase this sentence to make it more compatible with terminology, suggested by Richter et al.
Legend to Fig.4: Pln abbreviation is missing. Also, the abbreviation “vn” is a vellum nerve according to the legend, but on the corresponding picture “vn” points to a neuron (or at least it looks like)!
The true discussion begins from the line 568. All the above fragments look like a part of introduction to another paper since it consists of a review of the literature on various neurotransmitters without any discussion of the results. A logical question arises: how do these data compare with the results obtained by the authors? For what purpose are they presented here? In the presented form, the removal of this fragment will not fundamentally change anything in the text.
The last paragraph of the discussion also lacks a conclusion and thus looks incomplete.
Other minor comments:
Line 98: maiN morphological characters
Line 175: based
Line 178: Callista brevisiphonata should be italicized.
Line 184: oF the commissure
Line 224: The species name should be italicized. The same is for line 276, 323, 357, 394, and 480.
Line 502: The cerebral ganglia send, not sends.
Figure 9: What is “ens” and “f”?
Author Response
Dear Reviewer!
Thank you so much for the time you have spent on reviewing our text and all the comments you have made here. We have corrected all the comments you indicated, both textual and corrections in Figures and Fig. legends.
The paper by Nikishenko et al. represents an original descriptive and comparative study of the nervous system in different bivalve species at the pediveliger stage. The research is interesting with a clear and simple design. I really enjoyed reading it and would recommend for publication. However, I found some points that must be improved before publishing. Below you will find a list of my comments.
Legend for figure 1: abbreviation list is missing.
Reply
Dear Reviewer. Sorry for that. We have added the abbreviation for Fig1.
Dear
The figures themselves need to be improved. Sometimes it is not easy to find the structures, mentioned in the text. The abbreviations help a lot, but they are mostly about central nervous system and only few of them describe the PNS. Please, if it is not too hard, provide a bit more abbreviations helping to assess the details of the peripheral nervous system.
Reply
WE have added a separate schematic drawing (with two axes) of the pediveliger with all the designations to simplify the understanding of the complex structure of the nervous system of larvae
The lists of abbreviations in the legends are arranged in a non-alphabetic order. This makes them difficult to find.
Reply
Changed
Legend to Figure 5: What is ag – apical ganglion? Should it be the Apical organ? Please correct it here as well as on the other legends. Also, there is no “dn” and “pn” on the images. The same is for the legend for figure 6. It looks like a copy-paste artefact.
Reply
Corrected
Line 253: The word “morphostructure” looks a bit confusing, as far as I know this term is referred rather to geology, than biology.
Reply
Agree. changed
Line 288: The term neuropil is used a bit incorrectly. I’m afraid that it is impossible to apply the term neuropil to the group of neurites of the single cell. If my memory doesn’t let me down, the neuropil of the apical organ is called “apical neuropil”. Do you mean that the neurites of these cells contribute to it? To be honest, I think I understand what do you mean, but please, rephrase this sentence to make it more compatible with terminology, suggested by Richter et al.
Reply
Not really. Neuropil is the term used to define the dense network of fine glial and neuronal processes (enurites for invertebrates), and fibrils in the gray matter of CNS (for vertebrates). See gross anatomy of adult ganglia of mussels and scallop
Kniazkina, M., & Dyachuk, V. (2022). Neurogenesis of the scallop Azumapecten farreri: from the first larval sensory neurons to the definitive nervous system of juveniles. Frontiers in Zoology, 19(1), 1-19;
Kotsyuba, E. Distribution of molecules related to neurotransmission in the nervous system of the mussel Crenomytilus grayanus / E. Kotsyuba et al. // Frontiers in neuroanatomy. – 2020. – Т. 14. – С. 35).
Larvae of the late stages of development of bivalve mollusks have dense clusters of neurites (neuropiles), and each ganglion of the central nervous system of the veliger has its own neuropile, which may include neurons of different trasnmitter nature (see additional data).
Legend to Fig.4: Pln abbreviation is missing. Also, the abbreviation “vn” is a vellum nerve according to the legend, but on the corresponding picture “vn” points to a neuron (or at least it looks like)!
Reply
Thank you. Actually, this is velum neurites. Changed
The true discussion begins from the line 568. All the above fragments look like a part of introduction to another paper since it consists of a review of the literature on various neurotransmitters without any discussion of the results. A logical question arises: how do these data compare with the results obtained by the authors? For what purpose are they presented here? In the presented form, the removal of this fragment will not fundamentally change anything in the text.
The last paragraph of the discussion also lacks a conclusion and thus looks incomplete.
Reply
We do not agree with this statement and believe that discussing the distribution of neurotransmitters in larvae is very important, since this is the basis for the entire description of neuromorphology and, consequently, a comparative description of the neural structures of larvae with different lifestyles and taxonomic affiliation. Therefore, the distribution of neurotransmitters in the central nervous system and peripheral nervous system is important for discussion and is not something separate from the main text.
Other minor comments:
Line 98: maiN morphological characters
Reply
Done
Line 175: based
Reply
Done
Line 178: Callista brevisiphonata should be italicized.
Reply
Done
Line 184: oF the commissure
Reply
Done
Line 224: The species name should be italicized. The same is for line 276, 323, 357, 394, and 480.
Reply
Done
Line 502: The cerebral ganglia send, not sends.
Reply
Done
Figure 9: What is “ens” and “f”?
Reply
ens, enteric nervous system
f, foot
Sorry, added
Dear Reviewer! We apologize for the numerous typos and thank you very much for pointing out our mistakes.
Best wishes,
Dr Vyacheslav Dyachuk and team
Reviewer 2 Report
This manuscript by Nikishchenko et al. describes in detail the nervous system of late (pediveliger) larvae in eight bivalve species using immunofluorescence. The authors choose eight species that live in four distinct ecological conditions (2 species/ecological group) and characterize both the central and peripheral nervous systems with anti-serotonin and anti-FMRF antibodies. Although the neuroanatomy of a few of the species considered in this study (Crassostea, Azumapecten Mactromeris) had been previously reported, this work adds new information on several species and considers them together and in the context of their ecological niche. The images are beautiful and of high quality, and the presentation of the larvae in both "lateral" and "umbo" views allow the reader to visualize the structure of the bivalve nervous system. Moreover, the comparison allowed the authors to define the general conservation of the bivalve neural organization, as well as the late persistence of bivalve apical organs together with the cerebral ganglia. Overall, the manuscript presents new and interesting research, and the methodology is sound and well described. However, I think that some changes are required to improve the clarity and completeness of the work before publication.
Major remarks
The introduction describes nicely both the diversity of bivalve molluscs and the structure of their nervous system. Towards the end of the introduction however, the authors say: “However, neither studies of pre- and post-metamorphosis (late) development stages nor investigations into the cellular mechanisms of rearrangement the larval mode of life in the juvenile period have been published to date”. I think this does not describe accurately what the work is addressing: from this sentence, it sounds like the authors are going to compare different developmental stages. Instead, I would focus on the comparison of different species with different ecological niches.
Moreover, in the last two paragraphs, some explanation is required for why the pediveliger stage was chosen, and maybe a short description of the stage.
In the methods section, for immunocytochemistry (lines 111-137) please specify whether all larvae were treated with the same protocol. If not, please list the differences (I see for example that larvae were incubated with primary antibody for 3-5 days, does this depend on the species or the antibody?)
In line 125: in this or previous studies, have any negative control been done to check for autofluorescence or aspecific binding? If so, please specify.
In the immunocytochemistry or in the confocal microscopy paragraphs, please provide some information on sample numbers: how many specimens were stained/checked/imaged?
The results section describes the bivalve neuromorphology in great details and is accompanied by beautiful immunofluorescence stainings. However, due to the complexity of mollusks’ neuroanatomy, it is sometimes difficult to follow the text descriptions in the figures, and I think that changes to both text and figures are required to improve clarity, especially since Biology is not a journal specialized in mollusks.
Figures:
- To help guide the reader on the figure, it would be important to add one or two morphological landmarks which are in common to all species (I think the umbo and anterior or posterior end would be ideal), so that the relative position of the different neural structures considered can be better understood across the different figures.
- To help compare the different species, please keep the orientation of the animals (in lateral view) the same throughout the manuscript. This, together with the morphological landmarks, will make it easier to navigate the figures.
- For similar reasons, it would be useful to have an indication of the reciprocal orientation between lateral (A,C) and umbo (B,D) views in each species (perhaps an arrow on the lateral view pointing in the direction of the umbo view?).
- Make sure to refer to all supplementary figures in the main text. Furthermore, in supplementary figures please make sure that the orientation of the specimens is the same as the one shown in the main figures.
- The figure legends need corrections: importantly, make sure that all the abbreviations are present in the figure. Also, make sure that all abbreviations and all symbols are explained in the figure legend. Please order the abbreviations alphabetically. Also, in all legends please add the larval stage shown, even if it’s always pediveliger.
- I strongly believe that a schematic summary figure (Figure 10) is required here to properly visualize the similarities and differences between species. This can be along the same lines of the beautiful Figure 9 from Kniazkina & Dyachuk, 2022 (Frontiers in Zoology 19:22) or Figure 5 from Yurchenko et al., 2019 (EvoDevo 10:22), but showing the differences among species. I think this figure will highlight the strength of this paper, i.e. the comparative description of neuroanatomy in multiple species.
Text:
- Because the neuroanatomy is very complex, it is sometimes difficult to follow the text and figures. I think it would be useful to mention in the text the abbreviations used in the figure (or at least the exact same terms) corresponding to the structures described. For example, in line 255-256, is the sentence referring to the aan? If so, why is it defined as a nerve net? And in line 257-259, where in Figure 3A are the pedal nerves?
- Please make sure that the absence of signal or the difference between species is also properly described in the results and not only in the discussion. For example, mention the fact that P. macrochisma has more than one stomatogastric neuron, and report when immunoreactivity for one of the markers is not detected in the stomatogastric neurons (eg. Serotonin in C. gigas and C. grayanus?).
In addition, the following edits are not required for publication in my opinion (and therefore do not need to be addressed during the revision), but if possible it would be nice to add them to the revised version of the manuscript.
- Is it possible to have an estimate of the number of 5-HT-positive neurons of the apical organ in different species? Can this be added to the results?
- ‘Species and criteria’ paragraph: I really appreciated the presence of the adult bivalves images, which give a good idea of the animal morphology and size differences. Since the species that are ced in this study show ecological differences, please describe in this section, based also on the beautiful pictures provided, if there are any obvious/large scale morphological differences (also known from previous literature) between ecological groups.
- The neurites detailed structures are sometimes difficult to discern clearly in the figures. If possible, please consider adding magnifications (in the figures or as supplementary materials) of particular regions (particularly for the CNS) where neurites can be visualized more clearly.
The discussion provides a comprehensive review of neurotransmitter distribution in bivalves. Moreover, the reported results are considered in the context of the previous literature, proposing some changes in the interpretation of neuroanatomical data. However, I think that two aspects of the discussion can be improved:
- While one of the key points of the introduction, which also justifies the structure in which the results are presented, is the difference in ecological features of the species studied, in the discussion only a few sentences are dedicated to this. While the conclusions are perfectly acceptable, please comment on this more in the discussion section, and maybe relate it to differences in other anatomical structures if they have been reported in the literature (perhaps this could be added after lines 575-577).
- This is related to the other aspect of the comparison that is should be expanded: the evolutionary considerations. These species belong to a variety of Bivalvia orders and families. Are some of the similarities and differences due to their evolutionary relationships? I would dedicate some space in the discussion for this, similar to lines 616-619, but more broadly, and I would comment on the possible influence of evolutionary history in addition to ecology. If the discussion becomes too long, I suggest condensing the initial portion on the review of neurotransmitters’ distribution, as it is quite extensive.
The references need major revisions, as several of the referenced articles have missing information (e.g. the publication year) or the author order is incorrect.
Minor corrections
Abstract: The abstract is generally clear; however I recommend adding one or two sentences at the beginning to give general information on bivalve nervous systems and introduce the scope of the work.
Line 30: “on the bottom”: I think it might be better to use more technical expressions such as “in benthic environments”.
Line 40: all these references refer to the bivalve nervous system. Please add or substitute with more general articles describing bivalve life cycle.
Line 37: “because some species can lead a life that includes partially features of different forms” change to “because the life of some species includes features of different ecological groups”.
Line 44: “but the mechanisms of its formation have common features”: change to “But common mechanisms that control its formation have been described (ref)”.
Line 44: add reference for this statement.
Line 53: in Yurchenko et al., 2019, these are defined as “lateral nerve cords”. If this is a typo, please correct, otherwise explain the change in the name.
Line 83: specify that neural morphologies, and not the whole body anatomy, are compared.
Line 95: “and also papers with descriptions of the shell morphology”: change to “and to previous descriptions of cell morphology”.
Line 98: change “maim” with “main”.
Line 113: the text refers to “direct immunocytochemical staining”, but then describes an indirect method using both primary and secondary antibodies. Please change to “indirect”.
Line 119: “incubated for a day at 4°C”: does this mean overnight or for 24 hours? Please specify.
Line 125: change “used in a several of other studies” to “used in several other studies”.
Line 568: change “Do date” with “To date”.
Line 635: I think it might be better to change “may indicate conditions of their habitat” with “may indicate adaptation to different conditions of their habitat”.
I think that the introduction and discussion sections require some language editing. Some possible corrections are described in the “minor corrections” section in "Suggestions for Authors", but a more thorough language proofreading is recommended.
Author Response
Dear Reviewer!
Thank you so much for the time you have spent on reviewing our text and all the comments you have made here. We have corrected all the comments you indicated, both textual and corrections in drawings and legends.
This manuscript by Nikishchenko et al. describes in detail the nervous system of late (pediveliger) larvae in eight bivalve species using immunofluorescence. The authors choose eight species that live in four distinct ecological conditions (2 species/ecological group) and characterize both the central and peripheral nervous systems with anti-serotonin and anti-FMRF antibodies. Although the neuroanatomy of a few of the species considered in this study (Crassostea, Azumapecten Mactromeris) had been previously reported, this work adds new information on several species and considers them together and in the context of their ecological niche. The images are beautiful and of high quality, and the presentation of the larvae in both "lateral" and "umbo" views allow the reader to visualize the structure of the bivalve nervous system. Moreover, the comparison allowed the authors to define the general conservation of the bivalve neural organization, as well as the late persistence of bivalve apical organs together with the cerebral ganglia. Overall, the manuscript presents new and interesting research, and the methodology is sound and well described. However, I think that some changes are required to improve the clarity and completeness of the work before publication.
Major remarks
The introduction describes nicely both the diversity of bivalve molluscs and the structure of their nervous system. Towards the end of the introduction however, the authors say: “However, neither studies of pre- and post-metamorphosis (late) development stages nor investigations into the cellular mechanisms of rearrangement the larval mode of life in the juvenile period have been published to date”. I think this does not describe accurately what the work is addressing: from this sentence, it sounds like the authors are going to compare different developmental stages. Instead, I would focus on the comparison of different species with different ecological niches.
Moreover, in the last two paragraphs, some explanation is required for why the pediveliger stage was chosen, and maybe a short description of the stage.
Reply
Dear reviewer! Thank you so much for your valuable comments. We agree with this comment and have made the necessary textual changes. Namely, we changed the end of the introduction part and focused on comparing pediveligers with different ecological niches, and not all stages of mollsuks larvae, and added a sentence about peliveligers, as you asked us.
In the methods section, for immunocytochemistry (lines 111-137) please specify whether all larvae were treated with the same protocol. If not, please list the differences (I see for example that larvae were incubated with primary antibody for 3-5 days, does this depend on the species or the antibody?)
In line 125: in this or previous studies, have any negative control been done to check for autofluorescence or aspecific binding? If so, please specify.
In the immunocytochemistry or in the confocal microscopy paragraphs, please provide some information on sample numbers: how many specimens were stained/checked/imaged?
Reply
We have declared that pediveligers of all species were treated with the same immunoprotocol. Incubation in primary antibodies took 3 days at +4C. We also added part (Antibodies) with detailed explanations about the primary antibodies used in our work: studies in which they were cited, manufacturers' statements about their action and their recommendations. We clarified that we used material incubated only with secondary antibodies, without primary antibodies, as a control to exclude a specific antibody binding and control for 1st Abs: the method of depletion of antibodies by its antigen, according to the manufacturer's protocol, but only for serotonin. It is impossible to do this for FMRFa peptide due to the lack of a synthetic antigen to date. All information is presented in the section “Antibodies” in In the Methods section of the MS.
The results section describes the bivalve neuromorphology in great details and is accompanied by beautiful immunofluorescence stainings. However, due to the complexity of mollusks’ neuroanatomy, it is sometimes difficult to follow the text descriptions in the figures, and I think that changes to both text and figures are required to improve clarity, especially since Biology is not a journal specialized in mollusks.
Reply
Dear reviewer! Thank you for this comment. We have made multiple changes in the text and added additional schemes with symbols and their interpretation, so that it would be easier for the reader to understand the numerous neurostructures in different mollusk larvae. In addition, we have presented evolutionary scheme of the studied species to understand which structures are present or absent in the larvae of different both ecogroups and taxonomic groups. We really hope that our changes and additional schemes have simplified the perception of the material and made it more accessible to a wide range of readers.
Figures:
- To help guide the reader on the figure, it would be important to add one or two morphological landmarks which are in common to all species (I think the umbo and anterior or posterior end would be ideal), so that the relative position of the different neural structures considered can be better understood across the different figures.
- To help compare the different species, please keep the orientation of the animals (in lateral view) the same throughout the manuscript. This, together with the morphological landmarks, will make it easier to navigate the figures.
- For similar reasons, it would be useful to have an indication of the reciprocal orientation between lateral (A,C) and umbo (B,D) views in each species (perhaps an arrow on the lateral view pointing in the direction of the umbo view?).
- Make sure to refer to all supplementary figures in the main text. Furthermore, in supplementary figures please make sure that the orientation of the specimens is the same as the one shown in the main figures.
- The figure legends need corrections: importantly, make sure that all the abbreviations are present in the figure. Also, make sure that all abbreviations and all symbols are explained in the figure legend. Please order the abbreviations alphabetically. Also, in all legends please add the larval stage shown, even if it’s always pediveliger.
- I strongly believe that a schematic summary figure (Figure 10) is required here to properly visualize the similarities and differences between species. This can be along the same lines of the beautiful Figure 9 from Kniazkina & Dyachuk, 2022 (Frontiers in Zoology 19:22) or Figure 5 from Yurchenko et al., 2019 (EvoDevo 10:22), but showing the differences among species. I think this figure will highlight the strength of this paper, i.e. the comparative description of neuroanatomy in multiple species.
Reply
We have added morphological landmarks to the figures with each species, that show the orientation of the larval body. For better understanding, these landmarks correspond to the landmarks in the figures in brightfield (Fig. 1): umbo, velum, anterior and posterior end. We have made it so that larvae of all species are oriented the same way in the figures: umbo on the left and anterior end on top. The legends were corrected to match the designations in the figures. We also added schematic representations of the nervous system of each species in the discussion part (Fig. 10).
Text:
- Because the neuroanatomy is very complex, it is sometimes difficult to follow the text and figures. I think it would be useful to mention in the text the abbreviations used in the figure (or at least the exact same terms) corresponding to the structures described. For example, in line 255-256, is the sentence referring to the aan? If so, why is it defined as a nerve net? And in line 257-259, where in Figure 3A are the pedal nerves?
- Please make sure that the absence of signal or the difference between species is also properly described in the results and not only in the discussion. For example, mention the fact that P. macrochisma has more than one stomatogastric neuron, and report when immunoreactivity for one of the markers is not detected in the stomatogastric neurons (eg. Serotonin in C. gigas and C. grayanus?).
Reply
We simplified the text by replacing the full names of neurostructures with abbreviations corresponding to the figures. We also corrected inconsistencies between the designations in the figures and the text by adding the designation pn (Fig. 3). We added mentions about the absence of FMRFamide and serotonin signal in ens, sgn in C. gigas, absence in sgn, ens, pn and pln in C. grayanus, absence of FMRFamide signal in sgn in P. macrochisma, absence of sgn and pln in M. yessoensis and A. farreri, no signal in pn and pln in K. japonica, no ens in C. brevisiphonata.
In addition, the following edits are not required for publication in my opinion (and therefore do not need to be addressed during the revision), but if possible it would be nice to add them to the revised version of the manuscript.
- Is it possible to have an estimate of the number of 5-HT-positive neurons of the apical organ in different species? Can this be added to the results?
Reply
Since at pediveliger stage of development there is a partial reduction of larval organs, including the apical organ, as well as a decrease in the size of its cells and their compaction, there are difficulties in detecting the nuclei of apical organ cells for counting them. At earlier developmental stages (trochophora and/or veliger) this is possible because the ratio of the apical organ size to the larval body is much smaller, as well as its density.
- ‘Species and criteria’ paragraph: I really appreciated the presence of the adult bivalves images, which give a good idea of the animal morphology and size differences. Since the species that are ced in this study show ecological differences, please describe in this section, based also on the beautiful pictures provided, if there are any obvious/large scale morphological differences (also known from previous literature) between ecological groups.
Reply
The part about ecological groups was added to the discussion: unfortunately, there are few unambiguous papers even on adult bivalves, much more papers on the dependence of the shell shape of bivalves on the lifestyle and substrate they inhabit. Before us, no one has tried to compare larval stages of bivalves in the context of ecological groups.
- The neurites detailed structures are sometimes difficult to discern clearly in the figures. If possible, please consider adding magnifications (in the figures or as supplementary materials) of particular regions (particularly for the CNS) where neurites can be visualized more clearly.
Reply
We supplemented the figures with neuromorphology of pediveligers of each species with magnified images of the CNS.
The discussion provides a comprehensive review of neurotransmitter distribution in bivalves. Moreover, the reported results are considered in the context of the previous literature, proposing some changes in the interpretation of neuroanatomical data. However, I think that two aspects of the discussion can be improved:
- While one of the key points of the introduction, which also justifies the structure in which the results are presented, is the difference in ecological features of the species studied, in the discussion only a few sentences are dedicated to this. While the conclusions are perfectly acceptable, please comment on this more in the discussion section, and maybe relate it to differences in other anatomical structures if they have been reported in the literature (perhaps this could be added after lines 575-577).
- This is related to the other aspect of the comparison that is should be expanded: the evolutionary considerations. These species belong to a variety of Bivalvia orders and families. Are some of the similarities and differences due to their evolutionary relationships? I would dedicate some space in the discussion for this, similar to lines 616-619, but more broadly, and I would comment on the possible influence of evolutionary history in addition to ecology. If the discussion becomes too long, I suggest condensing the initial portion on the review of neurotransmitters’ distribution, as it is quite extensive.
Reply
We expanded the discussion, added a part about ecological groups, and provided a schematic tree with the evolution of neurostructures in pediveligers of the bivalves we studied.
The references need major revisions, as several of the referenced articles have missing information (e.g. the publication year) or the author order is incorrect.
Minor corrections
Abstract: The abstract is generally clear; however I recommend adding one or two sentences at the beginning to give general information on bivalve nervous systems and introduce the scope of the work.
Reply
Changed.
Line 30: “on the bottom”: I think it might be better to use more technical expressions such as “in benthic environments”.
Reply
Changed.
Line 40: all these references refer to the bivalve nervous system. Please add or substitute with more general articles describing bivalve life cycle.
Reply
Added
Line 37: “because some species can lead a life that includes partially features of different forms” change to “because the life of some species includes features of different ecological groups”.
Reply
Done
Line 44: “but the mechanisms of its formation have common features”: change to “But common mechanisms that control its formation have been described (ref)”.
Reply
Done
Line 44: add reference for this statement.
Reply
Added
Line 53: in Yurchenko et al., 2019, these are defined as “lateral nerve cords”. If this is a typo, please correct, otherwise explain the change in the name.
Reply
Corrected.
Line 83: specify that neural morphologies, and not the whole body anatomy, are compared.
Reply
Done
Line 95: “and also papers with descriptions of the shell morphology”: change to “and to previous descriptions of cell morphology”.
Reply
Done
Line 98: change “maim” with “main”.
Reply
Done
Line 113: the text refers to “direct immunocytochemical staining”, but then describes an indirect method using both primary and secondary antibodies. Please change to “indirect”.
Reply
Sorry, done
Line 119: “incubated for a day at 4°C”: does this mean overnight or for 24 hours? Please specify.
Reply
Changed
Line 125: change “used in a several of other studies” to “used in several other studies”.
Reply
Changed
Line 568: change “Do date” with “To date”.
Reply
Changed
Line 635: I think it might be better to change “may indicate conditions of their habitat” with “may indicate adaptation to different conditions of their habitat”.
Reply
Changed
Comments on the Quality of English Language
I think that the introduction and discussion sections require some language editing. Some possible corrections are described in the “minor corrections” section in "Suggestions for Authors", but a more thorough language proofreading is recommended.
Reply
Thank you. We have check language proofreadinga and made corrections
Best wishes,
Dr Vyacheslav Dyachuk and team
Reviewer 3 Report
This study deals with comparative observation of the nervous system structure in pediveliger of bivalves with different modes of life in the adult stage. This is an excellent work using a confocal microscope and worthful for neurogenesis of bivalves.
In the manuscript, the accuracy of figures and the descriptions is very important. I hope the authors take great care to make sure readers do not get lost. I think that there are several improvements that should be made before publication.
VG is not shown in immunoreactive staining for serotonin of Figs.2-9 A & B. However there are descriptions about VG in lines 436, 496 and 515. How can I resolve this discrepancy?
Similarly PG is not shown in immunoreactive staining for serotonin of Figs.3 &5, though there is a description about PG in line 324.
The direction of umbo view in the figure does not seem to be fixed. Fig. 5B & D appear to be wider than the other Figs, and conversely Fig.9B & D appear to be less wide than the other, though any larvae appear to be about the same size.
No description of pallial neurons in Figs. 6 &8. Is it considered that pallial neurons are not characteristic in these species?
Lines 257-258, I am not sure if “A wide plexus of pedal nerves detected in the foot” because the foot is not shown in the figure.
Line 282, “cerebral ones” means cerebral organs?
Line 333, blue color, tubulin is not shown.
No CG in Fig. 6A & B.
Figs. 6 & 7, I do not understand what the arrowheads indicate.
Lines 414-417, I fail to follow the explanation. It should be clarified what "the latter" refers to.
Lines 495-497, It is difficult to follow that CG is connected to VG because VG is not indicated.
Line 516, I fail to understand “this” ganglion.
Author Response
Dear Reviewer!
Thank you so much for the time you have spent on reviewing our text and all the comments you have made here. We have corrected all the comments you indicated, both textual and corrections in Figures and Fig. legends.
This study deals with comparative observation of the nervous system structure in pediveliger of bivalves with different modes of life in the adult stage. This is an excellent work using a confocal microscope and worthful for neurogenesis of bivalves.
In the manuscript, the accuracy of figures and the descriptions is very important. I hope the authors take great care to make sure readers do not get lost. I think that there are several improvements that should be made before publication.
VG is not shown in immunoreactive staining for serotonin of Figs.2-9 A & B. However, there are descriptions about VG in lines 436, 496 and 515. How can I resolve this discrepancy?
Reply
Fixed: we changed “PNS” (peripheral nervous system) to “VGc” (visceral ganglia commissure) in Figs. 8-9
Similarly PG is not shown in immunoreactive staining for serotonin of Figs.3 &5, though there is a description about PG in line 324.
Reply
We’re deeply sorry for these mistakes in the text to Figs. 3 & 5. We have fixed them: we removed mention of pedal nerves for Fig. 3 and changed ”PG” to ”PlG” for Fig. 5.
The direction of umbo view in the figure does not seem to be fixed. Fig. 5B & D appear to be wider than the other Figs, and conversely Fig.9B & D appear to be less wide than the other, though any larvae appear to be about the same size.
Reply
We fixed orientations of larvae in Figs. 2, 3, 5, 9. Now larvae of all species are oriented in the same direction. It is also important to note that the body width of bivalve's pediveligers is a species-specific trait, which is why the width differs in different figures.
No description of pallial neurons in Figs. 6 &8. Is it considered that pallial neurons are not characteristic in these species?
Reply
We have added mention about absence of these structures in C. grayanus and M. yessoensis.
Lines 257-258, I am not sure if “A wide plexus of pedal nerves detected in the foot” because the foot is not shown in the figure.
Reply
Fixed: we’re sorry for this mistake because they weren’t pedal nerves, but velum nerves.
Line 282, “cerebral ones” means cerebral organs?
Reply
By this we meant cerebral ganglia. Fixed.
Line 333, blue color, tubulin is not shown.
Reply
In Pododesmus pediveliger, tubulin did not stain clearly presumably due to species-specific features.
No CG in Fig. 6A & B.
Reply
We noted in the text that CG is absent in 5HT immunostaining.
Figs. 6 & 7, I do not understand what the arrowheads indicate.
Reply
Fixed: we have removed arrows without designations.
Lines 414-417, I fail to follow the explanation. It should be clarified what "the latter" refers to.
Reply
By this we meant CG. Fixed.
Lines 495-497, It is difficult to follow that CG is connected to VG because VG is not indicated.
Reply
Fixed: we changed “PNS” (peripheral nervous system) to “VGc” (visceral ganglia commissure) in Figs. 8-9
Line 516, I fail to understand “this” ganglion.
Reply
We meant by this PlG. Fixed.
Best wishes,
Dr Vyacheslav Dyachuk and team
Reviewer 4 Report
This brief report provides original data on the comparative neuroanatomy of pediveliger larvae of various bivalves from the Sea of Japan. The MS is generally well written with appropriate data analyses and an interesting discussion. Nevertheless, it has a number of shortcomings which should be addressed by the authors in order to improve understanding of the ms.
-All scientific names must be italic throughout the text including references. Please see the text.

- Minor editing of English language required
Author Response
Dear Reviewer!
Thank you so much for the time you have spent on reviewing our text and all the comments you have made here. We have corrected all the comments you indicated, both textual and corrections in Figures and Fig. legends.
Thank you so much for your comments and corrections in the text. This greatly facilitated our work on the text. Thank you for improving the text of the article.
Thanks
Best wishes,
Dr Vyacheslav Dyachuk and team
Round 2
Reviewer 2 Report
This revised version of the manuscript by Nikishchenko et al., describing the nervous system of pediveliger larvae in eight bivalve species using immunofluorescence, has addressed all the comments I have made in my first review report. I think it now explains the data clearly, and the authors have added two beautiful figures in the Discussion section that summarise their results clearly, and that will be very useful for the comparative analysis of mollusks' nervous systems. Therefore, I believe the manuscript is ready for publication after final proofreading.
I only have one final comment: in line 221, in the figure legend of Figure 1, the authors added "The letter designations are as follows:" but then did not complete it. During proofreading please complete this section.